# SpaceGNN: Multi-Space Graph Neural Network for Node Anomaly Detection with Extremely Limited Labels

**Xiangyu Dong[1], Xingyi Zhang[3], Lei Chen[2], Mingxuan Yuan[2], Sibo Wang[1]***
[1]The Chinese University of Hong Kong; [2]Huawei Noah's Ark Lab; [3]MBZUAI
{xydong, swang}@se.cuhk.edu.hk, xingyi.zhang@mbzuai.ac.ae,
{lc.leichen, yuan.mingxuan}@huawei.com

## Abstract

Node Anomaly Detection (NAD) has gained significant attention in the deep learning community due to its diverse applications in real-world scenarios. Existing NAD methods primarily embed graphs within a single Euclidean space, while overlooking the potential of non-Euclidean spaces. Besides, to address the prevalent issue of limited supervision in real NAD tasks, previous methods tend to leverage synthetic data to collect auxiliary information, which is not an effective solution as shown in our experiments. To overcome these challenges, we introduce a novel SpaceGNN model designed for NAD tasks with extremely limited labels. Specifically, we provide deeper insights into a task-relevant framework by empirically analyzing the benefits of different spaces for node representations, based on which, we design a Learnable Space Projection function that effectively encodes nodes into suitable spaces. Besides, we introduce the concept of weighted homogeneity, which we empirically and theoretically validate as an effective coefficient during information propagation. This concept inspires the design of the Distance Aware Propagation module. Furthermore, we propose the Multiple Space Ensemble module, which extracts comprehensive information for NAD under conditions of extremely limited supervision. Our findings indicate that this module is more beneficial than data augmentation techniques for NAD. Extensive experiments conducted on 9 real datasets confirm the superiority of SpaceGNN, which outperforms the best rival by an average of 8.55% in AUC and 4.31% in F1 scores. Our code is available at https://github.com/xydong127/SpaceGNN.

## 1 Introduction

With the rapid development of the Internet in recent years, graph-structured data has become ubiquitous. However, this popularity also presents a significant challenge: identifying anomalous nodes within a graph to prevent them from compromising the entire system. This task is commonly known as Node Anomaly Detection (NAD), which appears in various real-world scenarios, such as detecting money laundering in financial networks (Huang et al., 2022), identifying malicious comments in review networks (Li et al., 2019), and spotting bots on social platforms (Guo et al., 2022). While NAD is crucial for maintaining the integrity of these systems, effectively addressing it presents several challenges. Firstly, graph data inherently captures complex relationships across various domains, and the intricate shapes of the data complicate the accurate generation of node representations for NAD (Dong et al., 2025). Secondly, the ubiquitous issue of limited supervision in real scenarios makes it even harder to obtain sufficient comprehensive information for various types of nodes.

In the literature, researchers have widely employed Graph Neural Networks (GNNs) in their methods to solve general graph-related tasks. They have explored multiple frameworks from different spaces to enhance the expressiveness of their GNNs. For instance, GCN (Kipf & Welling, 2017), GraphSage (Hamilton et al., 2017), GAT (Velickovic et al., 2018), and GIN (Xu et al., 2019) embed

---

*Sibo Wang is the corresponding author.

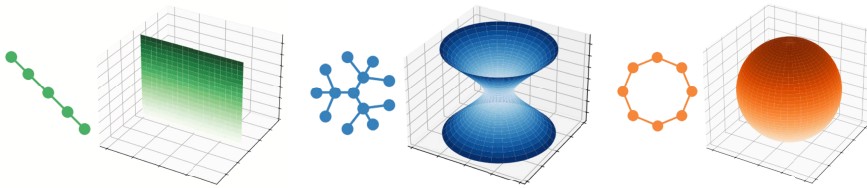

(a) Euclidean Space ($\kappa = 0$)     (b) Hyperbolic Space ($\kappa < 0$)     (c) Spherical Space ($\kappa > 0$)

Figure 1: Diverse data shapes with corresponding suitable projection spaces, where $\kappa$ represents the curvature of the space.

graphs into Euclidean space, while HNN (Ganea et al., 2018), HGCN (Chami et al., 2019), and HYLA (Yu & Sa, 2023) encode graphs into non-Euclidean space.

In addition to the generalized GNNs mentioned above, various specialized GNNs have been developed to address complex challenges in NAD tasks. Some of these models leverage the properties of anomalous nodes within Euclidean space to improve the quality of node representations for NAD, including GDN (Gao et al., 2023a), SparseGAD (Gong et al., 2023), GAGA (Wang et al., 2023b) and XGBGraph (Tang et al., 2023). Others enhance performance by employing a spectral view in Euclidean space, such as AMNet (Chai et al., 2022), BWGNN (Tang et al., 2022), and GHRN (Gao et al., 2023b). Moreover, to extract sufficient information from limited supervision, data augmentation techniques have been incorporated in recent work like CONSISGAD (Chen et al., 2024).

However, both generalized and specialized GNNs have overlooked the key challenges when applied to NAD tasks. For instance, without considering the diverse structural properties in real NAD scenarios, it is unlikely to design the most suitable node projection functions and propagation methods. As shown in Figure 1, we abstract subgraphs from real-world NAD datasets and provide their corresponding apt projection spaces (Bachmann et al., 2020). Specifically, the Euclidean space suits plain relational structures within graphs (Bandyopadhyay et al., 2020), serving as the base projection space for NAD tasks. However, certain subfields in NAD, such as rumor detection (Ma et al., 2018; Bian et al., 2020), require the ability to handle hierarchical data. The Hyperbolic space, which expands exponentially, is particularly adept at accommodating such data. Additionally, in financial networks, anomalies like money laundering crime usually display circle patterns (Dumitrescu et al., 2022; Altman et al., 2023). The Spherical space allows for a nuanced understanding of node properties in such data. As a result, directly encoding graphs from different NAD tasks in a single space with a fixed curvature $\kappa$ can result in suboptimal performance, as shown in our experiments. Furthermore, the limited supervision in real NAD applications presents another challenge for current methods. Given the imbalanced nature of the data, data augmentation techniques may not always effectively provide sufficient information, as will be shown in our experiments.

To address the above concerns, we present both empirical and theoretical analyses of NAD tasks with limited supervision. Motivated by this, we introduce SpaceGNN, a novel graph learning framework, which consists of three key components: **Learnable Space Projection (LSP)**, **Distance Aware Propagation (DAP)**, and **Multiple Space Ensemble (MulSE)**. Specifically, we design an insightful measure and conduct an empirical analysis to investigate the influence of various spaces on distinct classes of nodes, revealing the advantages of the adjustable projection function discussed in Section 4.2. Building on these empirical findings, we propose LSP as a projection function that embeds nodes into the most suitable spaces by a learnable curvature. Moreover, we introduce a novel metric to explore the benefits of a distance-based attention mechanism during propagation across different spaces. This metric further showcases the utility of various space representations from both empirical and theoretical perspectives, as elaborated in Section 4.3. Based on these results, we design DAP to adjust edge weights according to the distances within different spaces during feature propagation, which effectively mitigates the influence of noisy features propagated from different classes. Additionally, through an investigation of recent research, we empirically evaluate the limitations of relying on synthetic information via data augmentation in Section 4.4. To provide a more robust solution, we theoretically demonstrate that model augmentation approaches can serve as more effective alternatives. Thus, we develop MulSE to aggregate information from the ensemble of models from multiple spaces, a process shown to be effective from empirical and theoretical perspectives.

In summary, the main contributions of our work are as follows:

- We are the first, to the best of our knowledge, to reveal the benefits of leveraging multiple spaces for supervised NAD tasks from both empirical and theoretical perspectives.

- We propose SpaceGNN, a novel framework that ensembles comprehensive information from different spaces with a specialized attention mechanism based on solid analysis.

- Extensive experiments demonstrate the effectiveness of our framework. Compared to state-of-the-art models, SpaceGNN achieves superior performance in terms of AUC and F1 scores.

## 2 RELATED WORK

**Generalized GNNs.** GNNs have gained popularity for processing graph-structured data due to their outstanding ability to capture both structure and attribute information. For example, Euclidean GNNs, such as GCN (Kipf & Welling, 2017), extend convolution operations to graphs by aggregating information from neighboring nodes. Following this, GraphSAGE (Hamilton et al., 2017) introduces a sampling technique to enhance node representations. Subsequently, models like GAT (Velickovic et al., 2018) and GIN (Xu et al., 2019) are developed to increase the expressiveness of GNNs. In addition to Euclidean GNNs, non-Euclidean representations for nodes have drawn attention in the graph learning community due to their ability to encode special structures in graphs. For instance, HNN (Ganea et al., 2018) embeds node features into Hyperbolic space, which effectively captures hierarchical information. Based on this, HGCN (Chami et al., 2019) extends GNN principles to Hyperbolic space, leveraging its capability to represent hierarchical relationships more effectively. Recently, HYLA (Yu & Sa, 2023) combines Hyperbolic space with Laplacian-based graph learning, further enhancing the ability to capture hierarchical features in graph data. Despite the successes of generalized GNNs in some graph-related tasks, they often struggle to effectively address NAD problems due to their lack of consideration for specific properties within NAD tasks. In contrast, our proposed SpaceGNN leverages the combination of multi-space to collect comprehensive information, specifically targeting to solve the issues of NAD that generalized GNNs find difficult to process.

**Specialized GNNs.** Recognizing the limitations of generalized GNNs in NAD tasks, researchers have proposed several approaches specifically designed for this purpose. For instance, GDN (Gao et al., 2023a) is designed to resist high heterogeneity in anomalous nodes while benefiting the learning of normal nodes through homogeneity. Similarly, SparseGAD (Gong et al., 2023) introduces sparsity to the adjacency matrix to mitigate the negative influence of heterogeneity. GAGA (Wang et al., 2023b) employs a group aggregation technique to address low homogeneity issues. Additionally, XGBGraph (Tang et al., 2023) combines XGBoost with GIN to deal with tree-like data contained in NAD tasks. Beyond these spatial GNNs, several studies have explored NAD from a spectral view within Euclidean space. For example, AMNet (Chai et al., 2022) adaptively integrates signals of varying frequencies to extract more information. Following this, BWGNN (Tang et al., 2022) applies a Beta kernel to detect higher frequency anomalies. GHRN (Gao et al., 2023b) combines solutions from spectral space and homogeneity to boost performance in NAD tasks. Other than these studies, the most recent work, CONSISGAD (Chen et al., 2024), is proposed to tackle the limited supervision issue in NAD by generating pseudo labels. Despite the improvements in performance achieved by these specialized GNNs for NAD tasks, they lack a unified framework that integrates empirical and theoretical analysis related to NAD. This gap highlights the need for our proposed SpaceGNN, which aims to unify these valuable designs while providing a comprehensive understanding of their effectiveness in addressing NAD challenges.

## 3 PRELIMINARIES

A graph-structured data can be represented as $G = \{V, E, \boldsymbol{X}\}$, where $V$ is the node set, $E$ is the edge set, and $\boldsymbol{X} \in \mathbb{R}^{|V| \times d}$ is the node feature matrix. The $i$-th row $\boldsymbol{x}_i \in \mathbb{R}^d$ of $\boldsymbol{X}$ denotes the features of node $i \in V$. For a labeled node $i$, let $\boldsymbol{Y}_i \in \mathbb{R}^C$ denote the one-hot label vector, where $\boldsymbol{Y}_{ic} = 1$ if and only if node $i$ belongs to class $c$.

**Node Anomaly Detection (NAD).** NAD can be seen as a binary classification task, where nodes in the graph are categorized into two different categories: normal and anomalous. Specifically, $C = 2$ with label 0 representing normal class and label 1 representing anomalous class. Typically, the number of normal nodes is way larger than that of anomalous nodes, leading to an imbalanced dataset. Due to the sparse nature, there exists a thorny problem in the real NAD applications, e.g.,

the number of labeled nodes is extremely limited. Consequently, effectively leveraging the limited labels in datasets becomes a key challenge in NAD.

**Graph Neural Network (GNN).** A GNN consists of a sequence of fundamental operations, such as message passing through linear transformations and pointwise non-linear functions, which are performed on a set of nodes embedded in a given space. GNNs have been widely applied to various tasks related to graph-structured data. While these operations are well-understood in Euclidean space, extending them to non-Euclidean spaces presents challenges. After the concept of GNNs being generalized to operate on spaces with different curvature $\kappa$, allowing the network to be agnostic to the underlying geometry of the space. The propagation process for a node $i$ is as follows:

$$\boldsymbol{H}_i^{l+1} = \sigma(\exp_{\boldsymbol{x}'}^{\kappa}(\frac{1}{|N(i)|} \sum_{j \in N(i)} g_\theta(\log_{\boldsymbol{x}'}^{\kappa}(\boldsymbol{H}_i^l), \log_{\boldsymbol{x}'}^{\kappa}(\boldsymbol{H}_j^l)))),$$

where $\boldsymbol{H}_i^l$ is the $i$-th row of node representation matrix at the $l$-th layer, $\exp_{\boldsymbol{x}'}^{\kappa}(\cdot)$ and $\log_{\boldsymbol{x}'}^{\kappa}(\cdot)$ are projection functions specific to different spaces (Equations 1-2 show a possible choice for these two functions), $g_\theta(\cdot)$ is the aggregation function, $N(i)$ is the nodes within the one-hop neighborhood of node $i$, and $\sigma(\cdot)$ is the activation function. For GNNs in Euclidean space, where $\kappa = 0$, the projection functions act as identical mapping. In contrast, for GNNs operating in non-Euclidean space, where $\kappa \neq 0$ (as in (Chami et al., 2019)), two common choices for projection functions are the Poincaré Ball model and the Lorentz model, both characterized by $\kappa = -1$. Further details of these two models can be found in Appendix H.

**$\kappa$-stereographic model.** To further investigate the properties of NAD tasks across different spaces, the $\kappa$-stereographic model is introduced. This model can represent spaces with distinct curvature $\kappa$ that is not limited to $-1$. For a curvature $\kappa \in \mathbb{R}$ and a dimension $d \geq 2$, the model is defined as $M_\kappa^d = \{\boldsymbol{x} \in \mathbb{R}^d | -\kappa \|\boldsymbol{x}\|_2^2 < 1\}$. Note that when $\kappa \geq 0$, $M_\kappa^d$ is $\mathbb{R}^d$, while for $\kappa < 0$, $M_\kappa^d$ represents the open ball of radius $\frac{1}{\sqrt{-\kappa}}$. Following the extension presented by Bachmann et al. (2020), the $\kappa$-addition for $\boldsymbol{x}, \boldsymbol{y} \in M_\kappa^d$ is defined as:

$$\boldsymbol{x} \oplus_\kappa \boldsymbol{y} = \frac{(1 - 2\kappa\boldsymbol{x}^T\boldsymbol{y} - \kappa\|\boldsymbol{y}\|^2)\boldsymbol{x} + (1 + \kappa\|\boldsymbol{x}\|^2)\boldsymbol{y}}{1 - 2\kappa\boldsymbol{x}^T\boldsymbol{y} + \kappa^2\|\boldsymbol{x}\|^2\|\boldsymbol{y}\|^2} \in M_\kappa^d.$$

The projection functions $\exp_{\boldsymbol{x}'}^{\kappa}(\cdot)$ and $\log_{\boldsymbol{x}'}^{\kappa}(\cdot)$ can be defined as:

$$\exp_{\boldsymbol{x}'}^{\kappa}(\boldsymbol{x}) = \boldsymbol{x}' \oplus_\kappa (\tan_\kappa(|\kappa|^{\frac{1}{2}} \frac{\lambda_{\boldsymbol{x}'}^{\kappa}\|\boldsymbol{x}\|}{2}) \frac{\boldsymbol{x}}{\|\boldsymbol{x}\|}), \tag{1}$$

$$\log_{\boldsymbol{x}'}^{\kappa}(\boldsymbol{x}) = \frac{2|\kappa|^{-\frac{1}{2}}}{\lambda_{\boldsymbol{x}'}^{\kappa}} \tan_\kappa^{-1} \|(-\boldsymbol{x}') \oplus_\kappa \boldsymbol{x}\| \frac{(-\boldsymbol{x}') \oplus_\kappa \boldsymbol{x}}{\|(-\boldsymbol{x}') \oplus_\kappa \boldsymbol{x}\|}, \tag{2}$$

where $\boldsymbol{x}'$ can be chosen as the origin of the specific space, $\lambda_{\boldsymbol{x}'}^{\kappa} = \frac{2}{1+\kappa\|\boldsymbol{x}'\|^2}$, and $\tan_\kappa$ is the curvature-dependent trigonometric function defined as follows:

$$\tan_\kappa(\boldsymbol{x}) = \begin{cases} \frac{1}{\sqrt{-\kappa}} \tanh(\sqrt{-\kappa}\boldsymbol{x}), & \kappa < 0, \\ \boldsymbol{x}, & \kappa = 0, \\ \frac{1}{\sqrt{\kappa}} \tan(\sqrt{\kappa}\boldsymbol{x}), & \kappa > 0. \end{cases}$$

With the detailed definition of the $\kappa$-stereographic model established, a straightforward approach to design a GNN is to replace the projection functions by selecting a specific $\kappa$. However, this common design may not be suitable for NAD, as will be demonstrated in Section 4.2, where the usefulness of applying learnable projection functions to NAD will be highlighted.

# 4 OUR METHOD: SPACEGNN

## 4.1 OVERVIEW OF SPACEGNN

The proposed SpaceGNN consists of three key components, which will be discussed in detail in Sections 4.2-4.4. In Section 4.2, we begin by defining the expansion rate of different spaces and then utilize this concept to empirically demonstrate the advantages of incorporating learnable curvature

in NAD tasks, which serves as the foundation for designing the LSP module. Next, in Section 4.3, we revisit the concept of homogeneity and introduce weighted homogeneity to highlight the effectiveness of distance-aware attention across various spaces for NAD tasks. We also provide a theoretical analysis to justify the inclusion of distance-based propagation in diverse spaces, leading to the design of the DAP module. Finally, in Section 4.4, we first show the potential drawbacks of data augmentation techniques and then demonstrate the advantages of employing an ensemble of multiple spaces instead of relying on a single one. This motivates the design of the MulSE module, a robust solution for NAD tasks with limited supervision.

## 4.2 LEARNABLE SPACE PROJECTION

Before delving into the design of LSP, we first formulate the distance between two vectors $\boldsymbol{x}, \boldsymbol{y} \in M_\kappa^d$ as follows:

$$d_\kappa(\boldsymbol{x}, \boldsymbol{y}) = 2 \tan_\kappa^{-1} ||(-\boldsymbol{x}) \oplus_\kappa \boldsymbol{y}||. \tag{3}$$

Notice that this distance is not applicable when $\kappa = 0$. In such cases, we leverage the Euclidean distance, denoted as $d_0(\boldsymbol{x}, \boldsymbol{y})$.

Recap from Section 3 that NAD tasks can be viewed as binary classification tasks. Consider three data points: $\boldsymbol{x}_0$ and $\boldsymbol{x}_1$ belonging to class 0, and $\boldsymbol{y}$ belonging to class 1. An optimal model trained on such a dataset should minimize the distances between data points in the same class while maximizing the distances between data points from different classes. The ideal scenario is $d_\kappa(\boldsymbol{x}_0, \boldsymbol{x}_1) \approx 0$, $d_\kappa(\boldsymbol{x}_0, \boldsymbol{y}) \approx \infty$, and $d_\kappa(\boldsymbol{x}_1, \boldsymbol{y}) \approx \infty$. In such a case, even rule-based techniques can classify the data points correctly. To quantify the advantages of the projections in spaces with curvature $\kappa$, we propose a measure that reflects the scenario, which is stated in the following definition.

**Definition 1** (Expansion Rate). *For three data points, $\boldsymbol{x}_0$ and $\boldsymbol{x}_1$ in class 0 and $\boldsymbol{y}$ in class 1, let the inter-distance be $d_\kappa(\boldsymbol{x}_0, \boldsymbol{y})$ and intra-distance be $d_\kappa(\boldsymbol{x}_0, \boldsymbol{x}_1)$, then we can denote the ratio between them as $r_\kappa(\boldsymbol{x}_0, \boldsymbol{x}_1, \boldsymbol{y}) = \frac{d_\kappa(\boldsymbol{x}_0, \boldsymbol{y})}{d_\kappa(\boldsymbol{x}_0, \boldsymbol{x}_1)}$. Based on this, the Expansion Rate can be further defined as $ER_\kappa(\boldsymbol{x}_0, \boldsymbol{x}_1, \boldsymbol{y}) = \frac{r_\kappa(\boldsymbol{x}_0, \boldsymbol{x}_1, \boldsymbol{y})}{r_0(\boldsymbol{x}_0, \boldsymbol{x}_1, \boldsymbol{y})}$.*

As for the ratio $r_\kappa(\boldsymbol{x}_0, \boldsymbol{x}_1, \boldsymbol{y})$, an effective projection into space with curvature $\kappa$ should maximize the separation between data points from different classes, making their distances sufficiently distinct for accurate detection. Since data points are originally embedded in Euclidean space, it is natural to utilize the ratio in Euclidean space as the base to investigate the changes that occur when data points are projected into different spaces. Hence, the Expansion Rate quantifies the extent to which this ratio is expanded by the projection. To be specific, if $ER_\kappa(\boldsymbol{x}_0, \boldsymbol{x}_1, \boldsymbol{y}) > 1$, it indicates that the projection into the space with curvature $\kappa$ will have positive effects on NAD tasks, otherwise, it suggests that staying within Euclidean space may be more beneficial to the task.

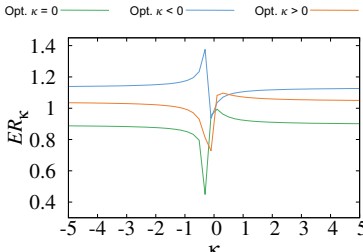

Figure 2: $ER_\kappa$ for different node triplets, varying based on $\kappa$. (Opt. stands for optimal.)

As shown in Figure 2, we investigate several node triplets within real datasets and plot the $ER_\kappa$ for representatives, varying based on $\kappa$. Specifically, for the blue line, the maximum value of $ER_\kappa$ is achieved when $\kappa$ is negative; for the orange line, the maximum value of $ER_\kappa$ is gained when $\kappa$ is positive; and for the green line, the maximum value of $ER_\kappa$ is obtained when $\kappa$ is 0. Based on these observations, it is desirable to design a GNN framework with learnable curvature, enabling the model to capture the optimal curvatures for nodes. To this end, we propose our base model architecture $f_{\boldsymbol{\kappa}}^L$ as follows:

$$\boldsymbol{E}^l = \text{CLAMP}_{\kappa^l}\left(\text{TRANS}(\exp_{\boldsymbol{o}}^{\kappa^l}(\boldsymbol{H}^l))\right),$$

$$\boldsymbol{H}_i^{l+1} = \phi(\log_{\boldsymbol{o}}^{\kappa^l}(\boldsymbol{E}_i^l) + \sum_{j \in N(i)} \omega_{ij}^{\kappa^l} \log_{\boldsymbol{o}}^{\kappa^l}(\boldsymbol{E}_j^l)), \tag{4}$$

$$\boldsymbol{Z} = \sigma(\text{MLP}(\text{CONCAT}(\boldsymbol{H}^0, \boldsymbol{H}^1, ..., \boldsymbol{H}^L))),$$

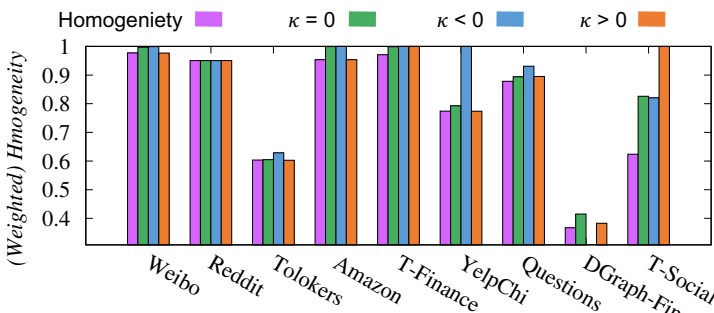

Figure 3: (Weighted) homogeneity of 9 real-world datasets.

where $\boldsymbol{H}^0 = \boldsymbol{X}$, TRANS($\cdot$) is the transformation function of the feature matrix built on two-layer linear projection and non-linear activation, CLAMP$_{\kappa^l}(\cdot)$ is the clamp function to restrict the node representations to a valid space, $\phi(\cdot)$ and $\sigma(\cdot)$ are the activation functions, $\exp_o^{\kappa^l}(\cdot)$ and $\log_o^{\kappa^l}(\cdot)$ are projection functions (Equations 1-2) based on the original point $\boldsymbol{o}$ of the corresponding space, $\omega_{ij}^{\kappa^l}$ is the coefficient based on the distances between node $i$ and $j$, which will be detailed in Section 4.3, and $\boldsymbol{Z} \in \mathbb{R}^{n \times 2}$ is the probability matrix under spaces with learnable curvatures $\boldsymbol{\kappa} \in \mathbb{R}^L$.

Based on the empirical analysis in this section, the importance of distances within different spaces stands out, which motivates us to further explore the effectiveness of incorporating the properties of these distances in our GNN framework. In the next subsection, we will first introduce the concept of weighted homogeneity and then provide both empirical and theoretical analysis to substantiate the rationale for designing a distance-aware attention mechanism for information propagation.

### 4.3 DISTANCE AWARE PROPAGATION

To elucidate the intuition behind weighted homogeneity, we first define the homogeneity for a node $i$ as $\frac{|\{j:j\in N(i), \boldsymbol{Y}_i = \boldsymbol{Y}_j\}|}{|N(i)|}$, which reflects the ratio of neighbors in the same category of $i$. Similarly, the homogeneity for a graph $G$ is defined as $\frac{\sum_{(i,j)\in E} \mathbb{I}[\boldsymbol{Y}_i = \boldsymbol{Y}_j]}{|E|}$, where $\mathbb{I}[\cdot]$ is the indicator function. This definition indicates the ratio of intra-edges within this graph. Hence, if we consider the information of each node as 1, the homogeneity metric can be interpreted as a measure of the information a node can gain from its neighbors with the same label during propagation. However, as will be demonstrated later, homogeneity alone is not an effective measure for guiding the message passing. Therefore, we introduce weighted homogeneity to enhance the passed information along intra-edges:

**Definition 2** (Weighted Homogeneity). *Let $\sigma$ denote the sigmoid function, $d_\kappa(\cdot, \cdot)$ denote the distance between two vectors in the space with curvature $\kappa$, then we can define the similarity vector for node $i$ as $\boldsymbol{s}_i^\kappa = \boldsymbol{1} - \sigma([d_\kappa(\boldsymbol{X}_i, \boldsymbol{X}_j) : j \in N(i)])$. The weighted homogeneity of a node $i$ can be defined as $WH_i^\kappa = \frac{\sum_{j\in N(i)} \boldsymbol{s}_{ij}^\kappa \mathbb{I}[\boldsymbol{Y}_i = \boldsymbol{Y}_j]}{\sum_{j\in N(i)} \boldsymbol{s}_{ij}^\kappa}$, and that of a graph $G$ can be defined as $WH^\kappa = \frac{\sum_{i\in V} \sum_{j\in N(i)} \boldsymbol{s}_{ij}^\kappa \mathbb{I}[\boldsymbol{Y}_i = \boldsymbol{Y}_j]}{\sum_{i\in V} \sum_{j\in N(i)} \boldsymbol{s}_{ij}^\kappa}$, where $\boldsymbol{s}_{ij}^\kappa$ is the $j$-th entry of $\boldsymbol{s}_i^\kappa$.*

In the above definition, $\boldsymbol{s}_i^\kappa$ represents the similarities between a node $i$ and its neighbors based on the distances of their representation in a space with curvature $\kappa$. This similarity vector is used as the weights in weighted homogeneity, allowing us to measure the information that a node can derive from its neighbors with the same label. If $\boldsymbol{s}_i^\kappa$ accurately reflects the similarities between nodes, it will assign larger weights to intra-edges, leading to the increased value of $WH^\kappa$. Its desirable benefits as coefficients in the propagation process will be shown in Theorem 1, that is, a higher $WH^\kappa$ correlates with improved performance of GNN. Next, we present a comparison between homogeneity and weighted homogeneity $WH^\kappa$ across 9 real-world datasets within different spaces to show the effectiveness of weighted homogeneity.

As shown in Figure 3, in most cases, $WH^\kappa$ with different $\kappa$ have higher values than homogeneity, which indicates that $WH^\kappa$ is beneficial to information from intra-edges. Besides, the results suggest that the optimal projection functions vary across different datasets, which further validates the motivation for integrating information from multiple spaces to enhance the performance of NAD.

Building on the above empirical findings, we present Theorem 1 to elucidate why $WH^\kappa$ serves as an effective measure for the propagation process. Detailed proof can be found in Appendix A.

**Theorem 1.** *Assume features of normal and anomalous nodes follow independent Gaussian distributions $\mathcal{N}(\boldsymbol{\mu}_n, \boldsymbol{\Sigma}_n)$ and $\mathcal{N}(\boldsymbol{\mu}_a, \boldsymbol{\Sigma}_a)$, respectively, and let $WH_\kappa$ (resp. $1 - WH_\kappa$) denote the coefficients of intra-edges (resp. inter-edges), then the probability of a node following its original distribution after a propagation process increases as $WH_\kappa$ increases.*

Theorem 1 emphasizes the importance of utilizing weighted homogeneity for effective information propagation. In particular, it mitigates the impact of noisy information passed from inter-edges while augmenting valuable information passed from intra-edges. Based on both empirical and theoretical results, we propose the DAP as follows:

$$\omega_{ij}^\kappa = \text{MLP}(\text{CONCAT}(\boldsymbol{X}_i, \hat{\boldsymbol{s}}_{ij}\boldsymbol{X}_j)), \tag{5}$$

where $\hat{\boldsymbol{s}}_{ij}^\kappa$ is the $j$-th entry of $\hat{\boldsymbol{s}}_i^\kappa$, which is an approximation of the similarity vector $\boldsymbol{s}_i^\kappa$ in Definition 2. The rationale behind approximating $\boldsymbol{s}_i^\kappa$ in DAP is to prevent the occurrence of invalid values for $d_\kappa(\cdot, \cdot)$ during the learning process. The details of this approximation are shown in Theorem 2:

**Theorem 2.** *Assume $\boldsymbol{x}, \boldsymbol{y} \in \mathbb{R}^d$ such that $\boldsymbol{x} \neq \boldsymbol{y}(\boldsymbol{x} \neq \frac{-\boldsymbol{y}}{\kappa\|\boldsymbol{y}\|^2}$ if $\kappa > 0)$, and $|\kappa| < \frac{1}{\min(\|\boldsymbol{x}\|^2, \|\boldsymbol{y}\|^2)}$, then $d_\kappa(\boldsymbol{x}, \boldsymbol{y}) \approx 2\|\boldsymbol{x} - \boldsymbol{y}\| - 2\kappa((\boldsymbol{x}^T\boldsymbol{y})\|\boldsymbol{x} - \boldsymbol{y}\|^2 + \frac{\|\boldsymbol{x}-\boldsymbol{y}\|^3}{3})$.*

The proof of Theorem 2 can be found in Appendix A. With this simple yet powerful design, our proposed SpaceGNN effectively harnesses information from intra-edges to boost the performance.

So far, we have presented the basic design of our model architecture $f_{\boldsymbol{\kappa}}^L$, which effectively captures information from spaces characterized by curvatures $\boldsymbol{\kappa} \in \mathbb{R}^L$. We have improved the expressiveness of GNN and explored properties underlying different spaces, tackling most problems existing in previous works. However, another thorny issue still remains: how to extract sufficient information when only extremely limited number of labels are available. In Section 4.4, we will empirically and theoretically demonstrate the advantages of utilizing an ensemble of GNNs from multiple spaces over data augmentation techniques for collecting auxiliary information in NAD with limited supervision.

## 4.4 MULTIPLE SPACE ENSEMBLE

In this section, we first provide an empirical analysis of CONSISGAD (Chen et al., 2024), the most recent work for the supervised NAD task aimed at tackling the issue of limited labels. CONSISGAD leverages a learnable framework to generate pseudo labels, thereby increasing the number of labeled nodes for training. However, the quality of these pseudo labels may not always be sufficient to provide valuable information. Moreover, as shown in a previous study (Wang et al., 2023a), the noise introduced by training samples with inaccurate pseudo labels will harm the final performance. Specifically, we investigate the well-trained CONSISGAD

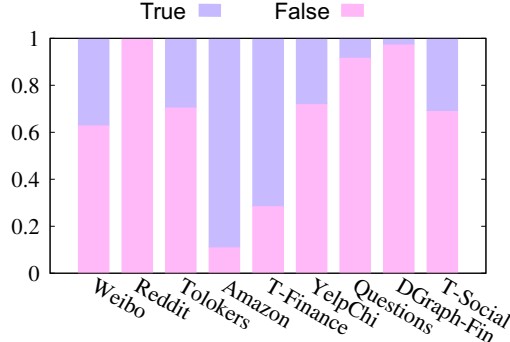

Figure 4: True/False Anomalous Rate.

framework and present the false rate of pseudo labels it generates on 9 real-world datasets in Figure 4. As we can observe, even with a fully-trained model, most generated anomalous labels are false across nearly all datasets. This situation negatively impacts the primary objective of NAD tasks, which is to detect anomalous nodes. Besides, due to the imbalanced nature of NAD tasks, the model tends to prioritize learning features of normal nodes. Consequently, with more normal labels and false anomalous labels, the model can finally degrade to a state of inferior performance.

Other than pseudo-label techniques, previous studies (Kirichenko et al., 2023; Lin et al., 2024) also point out the potential negative effects of other popular data augmentation techniques, such as increasing the bias of models and inducing the distribution shift between training and test data. Thus, we need to explore a more suitable way to tackle the issue of limited supervision existing in real NAD tasks. As shown in previous works (Xia et al., 2022; Liu et al., 2024), model augmentation can be an alternative way to enhance the limited information. The ensemble technique is one of the

most important ways to conduct model augmentation, as it brings enough benefits when there exist several independent views of the data. Such a characteristic provides explanations for combining our multiple GNNs within different spaces for NAD problems. Specifically, we have the following Proposition 1 to show the effectiveness of the ensemble of our multi-space framework on NAD tasks. The corresponding proof can be found in Appendix A.

**Proposition 1.** *Consider there exists a single node with label vector $\boldsymbol{p} \in \mathbb{R}^C$, and the corresponding probability vector $\boldsymbol{q}_i$ generated by our base model $f_{\boldsymbol{\kappa}_i}^L$, where $i = 1, \cdots, m$. Let $\bar{\boldsymbol{q}} = \sum_{i=1}^{m} \alpha_i \boldsymbol{q}_i$, where $\sum_{i=1}^{m} \alpha_i = 1$, and $\mathcal{L}(\cdot, \cdot)$ denote cross-entropy loss, then the ensemble cross-entropy loss, $\mathcal{L}(\boldsymbol{p}, \bar{\boldsymbol{q}})$, is upper bounded by weighted cross-entropy loss of $m$ single models, $\sum_{i=1}^{m} \alpha_i \mathcal{L}(\boldsymbol{p}, \boldsymbol{q}_i)$, with a gap term related to a label-dependent value $\Omega(\boldsymbol{p})$.*

Proposition 1 shows that, for every node in the graph, the combination of independent models of different spaces can reduce the loss during training, which can lead to better performance, indicating its superiority over the single-model framework. Further, building on the above Proposition 1, we provide the expected loss over graph $G$ in Proposition 2, whose proof can be found in Appendix A:

**Proposition 2.** *Let $\hat{\boldsymbol{q}}$ denote $\arg\min_{\boldsymbol{y} \in \boldsymbol{Y}} \mathbb{E}_G[\mathcal{L}(\boldsymbol{y}, \boldsymbol{q})]$, the centroid of model distribution with respect to $G$, then the ensemble cross-entropy loss over entire graph $G$, $\mathbb{E}_G[\mathcal{L}(\boldsymbol{p}, \bar{\boldsymbol{q}})]$, is upper bounded by weighted cross-entropy loss of $m$ single models, $\sum_{i=1}^{m} \alpha_i \mathcal{L}(\boldsymbol{p}, \hat{\boldsymbol{q}}_i)$, with a gap term related to a label-dependent value $\Theta(\boldsymbol{p})$.*

With the above empirical and theoretical analysis, we conclude that a safer and more effective way to tackle the limited supervision issues for supervised NAD tasks is the combination of views from multiple independent spaces, and thus we present our framework as follows:

$$f = \alpha f_{\boldsymbol{0}}^L + \sum_{i=1}^{H} \beta_i f_{\boldsymbol{\kappa}_i^-}^L + \sum_{j=1}^{S} \gamma_j f_{\boldsymbol{\kappa}_j^+}^L \tag{6}$$

where $f_{\boldsymbol{0}}^L$, $f_{\boldsymbol{\kappa}_i^-}^L$, and $f_{\boldsymbol{\kappa}_j^+}^L$ represent the Euclidean GNN, the $i$-th Hyperbolic GNN and the $j$-th Spherical GNN, respectively.

With empirical and theoretical analysis in Sections 4.2, 4.3, and 4.4, we present our SpaceGNN with a solid foundation, targeting to solve thorny issues in supervised NAD tasks, such as the existence of complex architectures, the requirement of the high-level expressive ability of the framework, and limited supervision. In the following Section 5, we will further provide the experimental results to show how effective our proposed framework is.

## 5 EXPERIMENTS

### 5.1 EXPERIMENTAL SETUP

**Datasets.** We evaluate SpaceGNN on 9 real-world datasets from the benchmark paper (Tang et al., 2023), including Weibo, Reddit, Tolokers, Amazon, T-Finance, YelpChi, Questions, DGraph-Fin, and T-Social. The detailed information on the datasets is listed in the Appendix B. To simulate the real application with limited supervision, we randomly divide each dataset into 50/50 for training/validation, and the rest of the nodes for testing.

**Baselines.** We compare SpaceGNN against 16 SOTA competitors, including generalized models, aiming to solve general graph-related tasks, and specialized models, which are designed for NAD.

- Generalized Models: MLP (Rosenblatt, 1958), GCN (Kipf & Welling, 2017), GraphSAGE (Hamilton et al., 2017), GAT (Velickovic et al., 2018), GIN (Xu et al., 2019), HNN (Ganea et al., 2018), HGCN (Chami et al., 2019), and HYLA (Yu & Sa, 2023).
- Specialized Models: AMNet (Chai et al., 2022), BWGNN (Tang et al., 2022), GDN (Gao et al., 2023a), SparseGAD (Gong et al., 2023), GHRN (Gao et al., 2023b), GAGA (Wang et al., 2023b), XGBGraph (Tang et al., 2023), and CONSISGAD (Chen et al., 2024).

**Experimental Settings**. To ensure a fair comparison, we obtain the source code of all competitors from GitHub and execute these models using the default parameter settings suggested by their authors. The hyperparameters of SpaceGNN are set according to the best value of the F1 score of the validation set in each dataset. The specific hyperparameters can be found in Appendix D.

Table 1: AUC and F1 scores (%) on 9 datasets with random split, compared with generalized models, where OOM represents out-of-memory.

| Datasets | Metrics | MLP | GCN | SAGE | GAT | GIN | HNN | HGCN | HYLA | SpaceGNN |
|---|---|---|---|---|---|---|---|---|---|---|
| Weibo | AUC | 0.4989 | 0.7338 | 0.6949 | 0.6887 | 0.4983 | 0.6726 | 0.8104 | 0.9056 | **0.9389** |
|  | F1 | 0.6393 | 0.6253 | 0.4399 | 0.4671 | 0.5404 | 0.6029 | 0.6773 | 0.5884 | **0.8541** |
| Reddit | AUC | 0.5892 | 0.5869 | 0.4074 | 0.5181 | 0.5954 | 0.5364 | 0.5348 | 0.4686 | **0.6159** |
|  | F1 | 0.4909 | 0.4820 | 0.4915 | 0.4915 | 0.4915 | 0.4915 | 0.4915 | 0.4915 | **0.4915** |
| Tolokers | AUC | 0.6462 | 0.6403 | 0.6786 | 0.5895 | 0.7004 | 0.6521 | 0.5980 | 0.5577 | **0.7089** |
|  | F1 | 0.4674 | 0.5697 | 0.5653 | 0.5526 | 0.5887 | 0.5446 | 0.5249 | 0.4888 | **0.6012** |
| Amazon | AUC | 0.8499 | 0.7677 | 0.7372 | 0.7487 | 0.7203 | 0.8597 | 0.7799 | 0.7192 | **0.9331** |
|  | F1 | 0.8441 | 0.3643 | 0.6339 | 0.4847 | 0.4822 | 0.6640 | 0.5608 | 0.5414 | **0.8935** |
| T-Finance | AUC | 0.8932 | 0.8882 | 0.6115 | 0.7210 | 0.8034 | 0.8773 | 0.9333 | 0.3935 | **0.9400** |
|  | F1 | 0.8354 | 0.7230 | 0.5740 | 0.6426 | 0.7313 | 0.8337 | 0.8656 | 0.4883 | **0.8723** |
| YelpChi | AUC | 0.5876 | 0.5385 | 0.5844 | 0.6120 | 0.5520 | 0.6519 | 0.5551 | 0.5505 | **0.6566** |
|  | F1 | 0.4623 | 0.4727 | 0.3711 | 0.4608 | 0.4608 | 0.5552 | 0.5029 | 0.4608 | **0.5719** |
| Questions | AUC | 0.4871 | 0.6075 | 0.5185 | 0.5002 | 0.5163 | 0.5073 | 0.5216 | 0.4052 | **0.6510** |
|  | F1 | 0.4984 | 0.4617 | 0.5041 | 0.4923 | 0.5045 | 0.4924 | 0.4965 | 0.4924 | **0.5336** |
| DGraph-Fin | AUC | 0.3830 | 0.4486 | 0.4127 | 0.3593 | 0.3781 | 0.3254 | 0.3299 | OOM | **0.6548** |
|  | F1 | 0.4608 | 0.3920 | 0.3842 | 0.4724 | 0.3790 | 0.4968 | 0.3312 | OOM | **0.5017** |
| T-Social | AUC | 0.5423 | 0.7523 | 0.6851 | 0.3593 | 0.7286 | 0.4728 | 0.4341 | OOM | **0.9392** |
|  | F1 | 0.4864 | 0.4473 | 0.5688 | 0.4724 | 0.5130 | 0.4923 | 0.4923 | OOM | **0.7571** |

## 5.2 EXPERIMENTAL RESULTS

We evaluate the performance of SpaceGNN against 8 generalized models and 8 specialized models. Tables 1 and 2 report the AUC and F1 scores of each model on 9 datasets, respectively. The best result on each dataset is highlighted in boldface. As we can see, SpaceGNN outperforms almost all baseline models on all datasets. Next, we provide our detailed observations.

Firstly, the most simple neural network, MLP, can only process the node features without considering structure information in the graph. The result is surprisingly high in several datasets compared with some generalized GNN models, which shows without correctly dealing with special structural properties, propagation may hurt performance in NAD tasks. In contrast, our SpaceGNN leverages information from multiple spaces, which is suitable for various structures, outperforming MLP by 17.34% and 9.91% on these 9 datasets in terms of average AUC and F1 scores, respectively.

Secondly, we examine four Euclidean GNNs, i.e., GCN, GraphSage, GAT, and GIN. They are the most popular GNNs for graph-related tasks. However, due to the lack of ability to tackle complex structures in different NAD datasets under limited supervision, they fail to generalize their power to such tasks, leading to inferior performance. In particular, compared with them, SpaceGNN takes the lead by 11.94%, 18.98%, 21.57%, and 17.17% on these 9 datasets in terms of average AUC score, and 17.10%, 17.16%, 17.12%, and 15.39% in terms of average F1 score, separately.

Thirdly, to the best of our knowledge, HNN is the first neural network that projects features into non-Euclidean space. The result aligns with the performance of MLP, that is, without carefully considering the suitable space for corresponding structures in NAD datasets, no structural information included may enhance the performance. On the contrary, our proposed model considers both Euclidean and non-Euclidean spaces for node projection and propagation, effectively utilizing structural features within graphs, so SpaceGNN can surpass HNN by 9.82% and 9.05% on these 9 datasets in terms of average AUC and F1 scores, respectively.

Fourthly, HGCN and HYLA are two non-Euclidean GNNs, encoding graph information into a single non-Euclidean space. The underfitting issue of complicated structural information in NAD graphs results in unsatisfactory performance on several datasets. In comparison with them, SpaceGNN captures enough features accurately through multiple spaces, exceeding them by 17.13% and 20.63% in terms of average AUC score, and 12.60% and 18.09% in terms of average F1 score, separately.

Fifthly, we compare our SpaceGNN with four specialized models within Euclidean space, i.e. GDN, SparseGAD, GAGA, and XGBGraph. They are built on the observation of the underlying properties of NAD graphs. Nevertheless, under limited supervision, the observed properties lose their power to be applied to the entire dataset. As a result, their performance degrades severely. In comparison, without manually detecting the special structures, our SpaceGNN can automatically capture accurate

Table 2: AUC and F1 scores (%) on 9 datasets with random split, compared with specialized models, where TLE represents the experiment can not be conducted successfully within 72 hours.

| Datasets | Metrics | AMNet | BWGNN | GDN | SparseGAD | GHRN | GAGA | XGBGraph | CONSISGAD | SpaceGNN |
|---|---|---|---|---|---|---|---|---|---|---|
| Weibo | AUC | 0.6367 | 0.7734 | 0.6392 | 0.6561 | 0.7722 | 0.6986 | 0.8421 | 0.7195 | **0.9389** |
| | F1 | 0.6560 | 0.7528 | 0.2516 | 0.6428 | 0.7344 | 0.6219 | 0.6353 | 0.6379 | **0.8541** |
| Reddit | AUC | 0.4024 | 0.5540 | 0.5716 | 0.4937 | 0.5583 | 0.4893 | 0.4768 | 0.4987 | **0.6159** |
| | F1 | 0.4915 | 0.4915 | 0.4915 | 0.4915 | 0.4915 | 0.4915 | 0.4944 | 0.4915 | **0.4915** |
| Tolokers | AUC | 0.5395 | 0.6202 | 0.6941 | 0.6739 | 0.6499 | 0.6137 | 0.6253 | 0.6531 | **0.7089** |
| | F1 | 0.5141 | 0.5698 | 0.5836 | 0.5325 | 0.5670 | 0.4889 | 0.5025 | 0.5652 | **0.6012** |
| Amazon | AUC | 0.9160 | 0.8661 | 0.8508 | 0.8698 | 0.8649 | 0.7772 | 0.8928 | 0.8993 | **0.9331** |
| | F1 | 0.7355 | 0.9006 | 0.6173 | 0.6016 | 0.8745 | 0.6509 | **0.9231** | 0.9031 | 0.8935 |
| T-Finance | AUC | 0.7885 | 0.8492 | 0.7200 | 0.6892 | 0.8758 | 0.8720 | 0.8563 | 0.9237 | **0.9400** |
| | F1 | 0.7576 | 0.6761 | 0.5339 | 0.2747 | 0.7777 | 0.7077 | 0.7673 | 0.8583 | **0.8723** |
| YelpChi | AUC | 0.5786 | 0.6012 | 0.5809 | 0.5652 | 0.6007 | 0.5351 | 0.5953 | 0.5983 | **0.6566** |
| | F1 | 0.4822 | 0.4608 | 0.5202 | 0.4608 | 0.4610 | 0.4868 | 0.5134 | 0.5133 | **0.5719** |
| Questions | AUC | 0.3628 | 0.5146 | 0.5044 | 0.5202 | 0.5118 | 0.5868 | 0.5069 | 0.6291 | **0.6510** |
| | F1 | 0.4924 | 0.5093 | 0.4950 | 0.4952 | 0.4934 | 0.4924 | 0.4548 | 0.5077 | **0.5336** |
| DGraph-Fin | AUC | 0.4359 | 0.4608 | 0.4723 | 0.3559 | 0.4216 | TLE | 0.4962 | 0.4345 | **0.6548** |
| | F1 | 0.4860 | 0.4975 | 0.4970 | 0.4988 | 0.4942 | TLE | 0.4947 | **0.5084** | 0.5017 |
| T-Social | AUC | 0.5239 | 0.6517 | 0.6259 | 0.5418 | 0.6336 | TLE | 0.7077 | 0.9129 | **0.9392** |
| | F1 | 0.5034 | 0.5204 | 0.5331 | 0.4923 | 0.5060 | TLE | 0.5604 | 0.7033 | **0.7571** |

properties underlying datasets in different spaces, taking the lead by 15.32%, 18.58%, 12.45%, and 11.54% in terms of average AUC score, and 17.26%, 17.63%, 12.54%, and 8.12% in terms of average F1 score, respectively.

Sixly, several specialized models, like AMNet, BWGNN, and GHRN, have considered NAD from a spectral view within Euclidean space, trying to gain diverse features to handle NAD tasks. Although such spectral kernels can detect frequency change because of anomalous nodes, they still fall behind our SpaceGNN by 20.60%, 12.75%, and 12.77% in terms of average AUC score, and 10.65%, 7.76%, and 7.52% in terms of average F1 score, separately, due to the inability to capture comprehensive enough properties within NAD datasets.

Finally, CONSISGAD tries to handle the problem of limited labels existing within real NAD tasks. Specifically, CONSISGAD, the most recent work in this area, generates pseudo labels for nodes to gain more supervision. However, as shown in Section 4.4, such a technique may lead to enormous noise, leading to subordinate results. By contrast, our theoretical analysis supports the usage of an ensemble of multiple spaces in SpaceGNN, and its performance demonstrates it empirically. To be specific, on 9 real-world datasets, CONSISGAD falls back SpaceGNN by 8.55% and 4.31% in terms of AUC and F1 scores on average, respectively.

Beyond the above results, we also provide parameter analysis, ablation study, additional experiments on different settings, an alternative model, the learned $\kappa$, time complexity, performance with more training data, and performance under GADBench (Tang et al., 2023) semi-supervised setting in Appendix E, F, G, H, I, J, K, and L, separately, which can further demonstrate the effectiveness of our proposed SpaceGNN.

## 6 CONCLUSION

In this paper, we provide detailed studies of the benefits of including multiple-space information to solve NAD tasks from both empirical and theoretical perspectives. Based on the results, we design SpaceGNN, an ensemble of diverse GNNs within Euclidean and non-Euclidean spaces, that can effectively handle difficulties in real NAD tasks, such as the appearance of complicated architectures, the need for powerful models, and limited labels. Extensive experiments demonstrate that SpaceGNN consistently outperforms other SOTA competitors by a significant margin, demonstrating our proposed SpaceGNN is an effective identifier for NAD tasks.

## ACKNOWLEDGMENTS

This work is supported by the RGC GRF grant (No. 14217322), Hong Kong ITC ITF grant (No. MRP/071/20X), and Tencent Rhino-Bird Focused Research Grant.

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

APPENDIX

A  PROOFS

**Proof of Theorem 1.** Let $p$ denote $WH^\kappa$, then the information a normal node can gain within its neighborhood during a propagation process follows $\mathcal{N}(p\boldsymbol{\mu}_n + (1-p)\boldsymbol{\mu}_a, p^2\boldsymbol{\Sigma}_n + (1-p)^2\boldsymbol{\Sigma}_a)$ according to the linear properties of independent Gaussian variables.

Let $\boldsymbol{X}$ and $\boldsymbol{Y}$ denote the distribution of the normal node and the information over $\mathbb{R}^d$, respectively. We then use Fréchet inception distance (Heusel et al., 2017) to describe the distance between two distributions as follows:

$$F(\boldsymbol{X}, \boldsymbol{Y})^2 = (\inf_{\gamma \in \Gamma(\boldsymbol{X}, \boldsymbol{Y})} \int_{\mathbb{R}^d \times \mathbb{R}^d} ||\boldsymbol{x} - \boldsymbol{y}||^2 d\gamma(\boldsymbol{x}, \boldsymbol{y})),$$
$$= (\inf_{\gamma \in \Gamma(\boldsymbol{X}, \boldsymbol{Y})} \mathbb{E}_{(\boldsymbol{x}, \boldsymbol{y}) \sim \gamma}[||\boldsymbol{x} - \boldsymbol{y}||^2]),$$

where $\Gamma(\boldsymbol{X}, \boldsymbol{Y})$ is the set of all measures on $\mathbb{R}^d \times \mathbb{R}^d$ with marginals $\boldsymbol{X}$ and $\boldsymbol{Y}$ on the first and second factors, separately. Hence, we have the following equation:

$$\mathbb{E}_{(\boldsymbol{x}, \boldsymbol{y}) \sim \gamma}[||\boldsymbol{x} - \boldsymbol{y}||^2]$$
$$= \mathbb{E}_{(\tilde{\boldsymbol{x}}, \tilde{\boldsymbol{y}}) \sim \tilde{\gamma}}[||(\tilde{\boldsymbol{x}} + \boldsymbol{\mu}_{\boldsymbol{x}}) - (\tilde{\boldsymbol{y}} + \boldsymbol{\mu}_{\boldsymbol{y}})||^2]$$
$$= \mathbb{E}_{(\tilde{\boldsymbol{x}}, \tilde{\boldsymbol{y}}) \sim \tilde{\gamma}}[||\tilde{\boldsymbol{x}} - \tilde{\boldsymbol{y}}||^2 + ||\boldsymbol{\mu}_{\boldsymbol{x}} - \boldsymbol{\mu}_{\boldsymbol{y}}||^2 + 2\langle \tilde{\boldsymbol{x}} - \tilde{\boldsymbol{y}}, \boldsymbol{\mu}_{\boldsymbol{x}} - \boldsymbol{\mu}_{\boldsymbol{y}}\rangle]$$
$$= ||\boldsymbol{\mu}_{\boldsymbol{x}} - \boldsymbol{\mu}_{\boldsymbol{y}}||^2 + \mathbb{E}_{(\tilde{\boldsymbol{x}}, \tilde{\boldsymbol{y}}) \sim \tilde{\gamma}}[||\tilde{\boldsymbol{x}} - \tilde{\boldsymbol{y}}||^2]$$

where $\boldsymbol{\mu}_{\boldsymbol{x}}$ and $\boldsymbol{\mu}_{\boldsymbol{y}}$ represent the mean value of distributions $\boldsymbol{X}$ and $\boldsymbol{Y}$, and $\tilde{\boldsymbol{x}}$ and $\tilde{\boldsymbol{y}}$ represent vectors following distribution $\tilde{\boldsymbol{X}}$ and $\tilde{\boldsymbol{Y}}$, which have 0 mean value and the same variance value as $\boldsymbol{X}$ and $\boldsymbol{Y}$, respectively. Hence, the Fréchet inception distance can be decomposed as:

$$F(\boldsymbol{X}, \boldsymbol{Y})^2 = ||\boldsymbol{\mu}_{\boldsymbol{x}} - \boldsymbol{\mu}_{\boldsymbol{y}}||^2 + F(\tilde{\boldsymbol{X}}, \tilde{\boldsymbol{Y}})^2$$

This result shows the distance between the distribution of the normal node and the information is determined by two parts, the mean value and the variance value. Specifically, we can assume $\boldsymbol{\Sigma}_n \approx \boldsymbol{\Sigma}_a \approx c\boldsymbol{I}$ in real NAD tasks, where $c$ is a small constant, due to the independent similar behaviors of nodes in the same category. Thus, we have $F(\tilde{\boldsymbol{X}}, \tilde{\boldsymbol{Y}})^2 \approx 0$ and $F(\boldsymbol{X}, \boldsymbol{Y})^2 = ||\boldsymbol{\mu}_{\boldsymbol{x}} - \boldsymbol{\mu}_{\boldsymbol{y}}||^2$.

Then, we check the distance between mean values of $\boldsymbol{X}$ and $\boldsymbol{Y}$. Specifically, it can be written as:

$$||\boldsymbol{\mu}_{\boldsymbol{x}} - \boldsymbol{\mu}_{\boldsymbol{y}}||^2 = (1-p)^2||\boldsymbol{\mu}_n - \boldsymbol{\mu}_a||^2$$

which concludes that if $||\boldsymbol{\mu}_n - \boldsymbol{\mu}_a||^2$ remains the same, as $p$ increases, the distance between the distribution of the normal node and the information will decrease, and thus the probability of a normal node following its original distribution after a propagation process increases as $WH_\kappa$ increases.

The situation of an anomalous node can be analyzed accordingly. This solution concludes that weighted homogeneity can benefit the propagation procedure for NAD tasks. $\square$

**Proof of Theorem 2.** First, we apply Taylor expansion on $\tan_\kappa^{-1}(t)$ for a fixed $t$ when $\kappa \to 0^+$:

$$\tan_\kappa^{-1}(t) = \kappa^{-\frac{1}{2}}\tan(\kappa^{\frac{1}{2}}t)$$
$$= \kappa^{-\frac{1}{2}}(\kappa^{\frac{1}{2}}t + \kappa^{\frac{3}{2}}\frac{t^3}{3} + \mathcal{O}(\kappa^{\frac{5}{2}}))$$
$$= t + \kappa\frac{t^3}{3} + \mathcal{O}(\kappa^2)$$

When $\kappa \to 0^-$:

$$\tan_\kappa^{-1}(t) = (-\kappa)^{-\frac{1}{2}}\tanh((-\kappa)^{\frac{1}{2}}t)$$
$$= (-\kappa)^{-\frac{1}{2}}((-\kappa)^{\frac{1}{2}}t - (-\kappa)^{\frac{3}{2}}\frac{t^3}{3} + \mathcal{O}(\kappa^{\frac{5}{2}}))$$
$$= t + \kappa\frac{t^3}{3} + \mathcal{O}(\kappa^2)$$

3

When $\kappa \to 0$, we also have $\tan_\kappa^{-1}(t) = t - \kappa\frac{t^3}{3} + \mathcal{O}(\kappa^2)$. Hence, we conclude that near 0, $\tan_\kappa^{-1}(t) = t - \kappa\frac{t^3}{3} + \mathcal{O}(\kappa^2)$.

Then, we need to use the Tayler expansion for $||\cdot||$. Specifically, $||\boldsymbol{x}+\boldsymbol{o}|| = ||\boldsymbol{x}||+\langle\boldsymbol{x}, \boldsymbol{o}\rangle+\mathcal{O}(||\boldsymbol{o}||^2)$ when $\boldsymbol{o} \to \boldsymbol{0}$.

After that, we derive the Tayler expansion for $\boldsymbol{x} \oplus_\kappa \boldsymbol{y}$ when $\kappa$ near 0:

$$
\begin{aligned}
\boldsymbol{x} \oplus \boldsymbol{y} =& \frac{(1 - 2\kappa\boldsymbol{x}^T\boldsymbol{y} - \kappa||\boldsymbol{y}||^2)\boldsymbol{x} + (1 + \kappa||\boldsymbol{x}||^2)\boldsymbol{y}}{1 - 2\kappa\boldsymbol{x}^T\boldsymbol{y} + \kappa^2||\boldsymbol{x}||^2||\boldsymbol{y}||^2} \\
=& ((1 - 2\kappa\boldsymbol{x}^T\boldsymbol{y} - \kappa||\boldsymbol{y}||^2)\boldsymbol{x} + (1 + \kappa||\boldsymbol{x}||^2)\boldsymbol{y})(1 + 2\kappa\boldsymbol{x}^T\boldsymbol{y} + \mathcal{O}(\kappa^2)) \\
=& (1 - 2\kappa\boldsymbol{x}^T\boldsymbol{y} - \kappa||\boldsymbol{y}||^2)\boldsymbol{x} + (1 + \kappa||\boldsymbol{x}||^2)\boldsymbol{y} + 2\kappa\boldsymbol{x}^T\boldsymbol{y}(\boldsymbol{x} + \boldsymbol{y}) + \mathcal{O}(\kappa^2) \\
=& (1 - \kappa||\boldsymbol{y}||^2)\boldsymbol{x} + (1 + \kappa||\boldsymbol{x}||^2)\boldsymbol{y} + 2\kappa(\boldsymbol{x}^T\boldsymbol{y})\boldsymbol{y} + \mathcal{O}(\kappa^2) \\
=& \boldsymbol{x} + \boldsymbol{y} + \kappa(||\boldsymbol{x}||^2\boldsymbol{y} - ||\boldsymbol{y}||^2\boldsymbol{x} + 2(\boldsymbol{x}^T\boldsymbol{y})\boldsymbol{y}) + \mathcal{O}(\kappa^2)
\end{aligned}
$$

By combining the above three Tayler expansions and ignore $\mathcal{O}(\kappa^2)$, we have the following equation:

$$
\begin{aligned}
d_\kappa(\boldsymbol{x}, \boldsymbol{y}) =& 2\tan_\kappa^{-1}(||(-\boldsymbol{x}) \oplus_\kappa \boldsymbol{y}||) \\
=& 2(||\boldsymbol{x} - \boldsymbol{y}|| + \kappa((-\boldsymbol{x})^T\boldsymbol{y})||\boldsymbol{x} - \boldsymbol{y}||^2)(1 - \frac{\kappa}{3}(||\boldsymbol{x} - \boldsymbol{y}||^2)) \\
=& 2||\boldsymbol{x} - \boldsymbol{y}|| - 2\kappa((\boldsymbol{x}^T\boldsymbol{y})||\boldsymbol{x} - \boldsymbol{y}||^2 + \frac{||\boldsymbol{x} - \boldsymbol{y}||^3}{3})
\end{aligned}
$$

which concludes our theorem. $\square$

**Proof of Proposition 1.** We use the weighted cross-entropy loss of $m$ single models to subtract the ensemble cross-entropy loss:

$$
\begin{aligned}
\sum_{i=1}^m \alpha_i\mathcal{L}(\boldsymbol{p}, \boldsymbol{q}_i) - \mathcal{L}(\boldsymbol{p}, \bar{\boldsymbol{q}}) =& \sum_{c=1}^C \boldsymbol{p}^c \log \bar{\boldsymbol{q}}^c - \sum_{i=1}^m\sum_{c=1}^C \alpha_i\boldsymbol{p}^c \log \boldsymbol{q}_i^c \\
=& \sum_{c=1}^C \boldsymbol{p}^c \log \bar{\boldsymbol{q}}^c - \sum_{c=1}^C \boldsymbol{p}^c \log(\prod_i(\boldsymbol{q}_i^c)^{\alpha_i}) \\
=& \sum_{c=1}^C \boldsymbol{p}^c \log(\frac{\bar{\boldsymbol{q}}^c}{\prod_i(\boldsymbol{q}_i^c)^{\alpha_i}}) \\
=& \Omega(\boldsymbol{p})
\end{aligned}
$$

By applying the weighted AM–GM inequality, we have $\bar{\boldsymbol{q}} \geq \prod_i(\boldsymbol{q}_i^c)^{\alpha_i}$, which means the term inside the log function is greater or equal to 1, and so the $\Omega(\boldsymbol{p})$ is non-negative. Thus, it finishes the proof of the Proposition. $\square$

**Proof of Proposition 2.** We take the expectation of the equation in Proposition 1 over entire graph $G$ and apply KL bias-variance decomposition in previous study (Wood et al., 2023), then we have:

$$
\begin{aligned}
\mathbb{E}_G[\mathcal{L}(\boldsymbol{p}, \bar{\boldsymbol{q}})] =& \mathbb{E}_G[\sum_{i=1}^m \alpha_i\mathcal{L}(\boldsymbol{p}, \boldsymbol{q}_i)] - \mathbb{E}_G[\Omega(\boldsymbol{p})] \\
=& \sum_{i=1}^m \alpha_i\mathcal{L}(\boldsymbol{p}, \hat{\boldsymbol{q}}_i) + \sum_{i=1}^m \alpha_i\mathbb{E}_G[D_{\mathrm{KL}}(\hat{\boldsymbol{q}}_i||\boldsymbol{q}_i)] - \mathbb{E}_G[\Omega(\boldsymbol{p})] \\
=& \sum_{i=1}^m \alpha_i\mathcal{L}(\boldsymbol{p}, \hat{\boldsymbol{q}}_i) + \Theta(\boldsymbol{p}),
\end{aligned}
$$

where $\Theta(\boldsymbol{p})$ is demonstrated as non-negative in previous study (Wood et al., 2023), and thus this Proposition is proven. $\square$

Table 3: Statistics of 9 datasets including the number of nodes and edges, the number of normal and anomalous nodes, the ratio of anomalous labels, average degree, and the node feature dimension.

| Datasets | #Nodes | #Edges | #Normal | #Anomalous | #Anomalous Rate | Average Degree | #Feature |
|---|---|---|---|---|---|---|---|
| Weibo | 8,405 | 407,963 | 7,537 | 868 | 0.1033 | 48.54 | 400 |
| Reddit | 10,984 | 168,016 | 10,618 | 366 | 0.0333 | 15.30 | 64 |
| Tolokers | 11,758 | 519,000 | 9,192 | 2,566 | 0.2182 | 44.14 | 10 |
| Amazon | 11,944 | 4,398,392 | 11,123 | 821 | 0.0687 | 368.25 | 25 |
| T-Finance | 39,357 | 21,222,543 | 37,554 | 1,803 | 0.0458 | 539.23 | 10 |
| YelpChi | 45,954 | 3,846,979 | 39,277 | 6,677 | 0.1453 | 83.71 | 32 |
| Questions | 48,921 | 153,540 | 47,461 | 1,460 | 0.0298 | 3.14 | 301 |
| DGraph-Fin | 3,700,550 | 4,300,999 | 1,210,092 | 15,509 | 0.0127 | 1.16 | 17 |
| T-Social | 5,781,065 | 73,105,508 | 5,606,785 | 174,280 | 0.0301 | 12.65 | 10 |

## B    DATASETS AND BASELINES

**Datasets.** The datasets used in our experiments are from the most recent benchmark paper (Tang et al., 2023), according to which, Weibo, Reddit, Questions, and T-Social aim to detect anomalous accounts on social media, Tolokers, Amazon, and YelpChi are proposed for malicious comments detection in review platforms, and T-Finance and DGraph-Fin focus on fraud detection in financial networks. The statistics of these 9 real-world datasets are shown in Table 3.

**Baselines.** The first group is generalized models:

- MLP (Rosenblatt, 1958): A type of neural network with multiple layers of fully connected artificial neurons;
- GCN (Kipf & Welling, 2017): A type of GNN that leverages convolution function on a graph to propagate information within the neighborhood of each node;
- GraphSAGE (Hamilton et al., 2017): A type of GNN that uses sampling technique to aggregate features from the neighborhood;
- GAT (Velickovic et al., 2018): A type of GNN that adopts an attention mechanism to assign different importance to different nodes within the neighborhood of each node;
- GIN (Xu et al., 2019): A type of GNN that captures the properties of a graph while following graph isomorphism;
- HNN (Ganea et al., 2018): A type of neural network that projects data features into non-Euclidean space;
- HGCN (Chami et al., 2019): A type of GNN that embeds node representations into non-Euclidean space and propagates accordingly;
- HYLA (Yu & Sa, 2023): A type of GNN combines both laplacian characteristics within a graph and the information from non-Euclidean space.

The second group is specialized models:

- AMMNet (Chai et al., 2022): A method proposed to capture both low- and high-frequency spectral information to detect anomalies;
- BWGNN (Tang et al., 2022): A method designed to handle the 'right-shift' phenomenon of graph anomalies in spectral space;
- GDN (Gao et al., 2023a): A method that aims to learn information from a graph of the dependence relationships between sensors;
- SparseGAD (Gong et al., 2023): A method that leverages sparsification to mitigate the heterophily issues within the neighborhood of each node;
- GHRN (Gao et al., 2023b): A method that tackles the heterophily problem in the spectral space of graph anomaly detection;
- GAGA (Wang et al., 2023b): A method that uses group aggregation to reduce the influence of low homophily;
- XGBGraph (Tang et al., 2023): A method that combines XGB and GIN to boost the expressiveness;
- CONSISGAD (Chen et al., 2024): A method that applies a pseudo-label generation technique to solve the limited supervision problem;

## C  ALGORITHM

---

**Algorithm 1:** $exp_o^\kappa/log_o^\kappa$

---

**Input:** $G$
**Output:** $\boldsymbol{H}^\kappa$
1  $\hat{\boldsymbol{H}} \leftarrow$ NORMALIZE$(\boldsymbol{H})$;
2  **if** $\kappa < 0$ **then**
3  $\quad \left\lfloor \ \boldsymbol{H}^\kappa \leftarrow \frac{tanh(\sqrt{|\kappa|}\hat{\boldsymbol{H}})\boldsymbol{H}}{\sqrt{|\kappa|}\hat{\boldsymbol{H}}} \text{ if } exp_o^\kappa, \text{ else } \frac{arctanh(\sqrt{|\kappa|}\hat{\boldsymbol{H}})\boldsymbol{H}}{\sqrt{|\kappa|}\hat{\boldsymbol{H}}} \right.$;
4  **else if** $\kappa > 0$ **then**
5  $\quad \left\lfloor \ \boldsymbol{H}^\kappa \leftarrow \frac{tan(\sqrt{|\kappa|}\hat{\boldsymbol{H}})\boldsymbol{H}}{\sqrt{|\kappa|}\hat{\boldsymbol{H}}} \text{ if } exp_o^\kappa, \text{ else } \frac{arctan(\sqrt{|\kappa|}\hat{\boldsymbol{H}})\boldsymbol{H}}{\sqrt{|\kappa|}\hat{\boldsymbol{H}}} \right.$;
6  **else**
7  $\quad \left\lfloor \ \boldsymbol{H}^\kappa \leftarrow \boldsymbol{H} \right.$;
8  Return $\boldsymbol{H}^\kappa$;

---

**Algorithm 2:** $CLAMP_\kappa$

---

**Input:** $G$
**Output:** $\boldsymbol{H}^\kappa$
1  $\hat{\boldsymbol{H}} \leftarrow$ NORMALIZE$(\boldsymbol{H})$;
2  $\epsilon \leftarrow 1^{-8}$;
3  $\tau \leftarrow \frac{1-\epsilon}{\sqrt{|\kappa|}}$;
4  **for** $i = 1$ *to* $n$ **do**
5  $\quad$ **for** $j = 1$ *to* $d$ **do**
6  $\quad\quad$ **if** $\hat{H}_{ij} > \tau$ **then**
7  $\quad\quad\quad \left\lfloor \ H_{ij}^\kappa \leftarrow \frac{H_{ij}}{\tau\hat{H}_{ij}} \right.$
8  $\quad\quad$ **else**
9  $\quad\quad\quad \left\lfloor \ H_{ij}^\kappa \leftarrow H_{ij} \right.$

10  Return $\boldsymbol{H}^\kappa$;

---

**Algorithm 3:** $f_{\boldsymbol{\kappa}}^L$

---

**Input:** $G$
**Output:** $\boldsymbol{Z}^{\boldsymbol{\kappa}}$
1  $\boldsymbol{H}^0 \leftarrow \boldsymbol{X}$;
2  **for** $l = 0$ *to* $L - 1$ **do**
3  $\quad \boldsymbol{E}^l \leftarrow CLAMP_{\kappa^l}(\sigma(exp_o^{\kappa^l}(\text{MLP}(\sigma(\text{MLP}(\boldsymbol{H}^l))))))$;
4  $\quad$ **for** $i = 1$ *to* $n$ **do**
5  $\quad\quad \hat{\boldsymbol{s}}_i^{\kappa^l} \leftarrow \boldsymbol{1} - \sigma([2||\boldsymbol{E}_i^l - \boldsymbol{E}_j^l|| - 2\kappa^l(\boldsymbol{E}_i^l)^T\boldsymbol{E}_j^l||\boldsymbol{E}_i^l - \boldsymbol{E}_j^l||^2 + \frac{||\boldsymbol{E}_i^l - \boldsymbol{E}_j^l||^3}{3} : j \in N(i)])$;
6  $\quad\quad \omega_{ij}^{\kappa^l} \leftarrow \text{MLP}(\text{CONCAT}(\boldsymbol{E}_i^l, \hat{\boldsymbol{s}}_{ij}^{\kappa^l}\boldsymbol{E}_j^l))$;
7  $\quad\quad \boldsymbol{H}_i^{l+1} \leftarrow \text{SELU}(log_o^{\kappa^l}(\boldsymbol{E}_i^l) + \sum_{j \in N(i)} \omega_{ij}^{\kappa^l} log_o^{\kappa^l}(\boldsymbol{E}_j^l))$;
8  $\boldsymbol{Z}^{\boldsymbol{\kappa}} \leftarrow \text{SOFTMAX}(\text{MLP}(\text{CONCAT}(\boldsymbol{H}^0, \boldsymbol{H}^1, ..., \boldsymbol{H}^L)))$;
9  Return $\boldsymbol{Z}^{\boldsymbol{\kappa}}$;

---

**Algorithm 4:** SpaceGNN

---

**Input:** $G, L$
**Output:** $\boldsymbol{Z}$
1  $\boldsymbol{Z}^{\boldsymbol{\kappa}^+} \leftarrow f_{\boldsymbol{\kappa}^+}^L(G), \boldsymbol{Z}^{\boldsymbol{\kappa}^-} \leftarrow f_{\boldsymbol{\kappa}^-}^L(G), \boldsymbol{Z}^{\boldsymbol{0}} \leftarrow f_{\boldsymbol{0}}^L(G)$;
2  $\boldsymbol{Z} \leftarrow (1 - \beta)((1 - \alpha)\boldsymbol{Z}^{\boldsymbol{\kappa}^-} + \alpha\boldsymbol{Z}^{\boldsymbol{\kappa}^+}) + \beta\boldsymbol{Z}^{\boldsymbol{0}}$;
3  Return $\boldsymbol{Z}$;

Table 4: Hyperparameters of 9 datasets for experiments in Section 5.

| Datasets | Learning Rate | Hidden Dimension | Layer | Dropout | Batch Size | $\alpha$ | $\beta$ |
|---|---|---|---|---|---|---|---|
| Weibo | 0.001 | 128 | 6 | 0 | 50 | 0.5 | 1 |
| Reddit | 0.001 | 128 | 5 | 0.05 | 50 | 0 | 0 |
| Tolokers | 0.001 | 128 | 1 | 0.05 | 50 | 0 | 0.5 |
| Amazon | 0.001 | 128 | 3 | 0.05 | 50 | 0 | 0 |
| T-Finance | 0.001 | 128 | 3 | 0.1 | 50 | 1 | 1 |
| YelpChi | 0.0001 | 128 | 1 | 0.1 | 50 | 0 | 0 |
| Questions | 0.0001 | 128 | 6 | 0.1 | 50 | 0.5 | 0.5 |
| DGraph-Fin | 0.001 | 128 | 4 | 0.05 | 50 | 0 | 1 |
| T-Social | 0.001 | 128 | 6 | 0.05 | 50 | 0.5 | 1 |

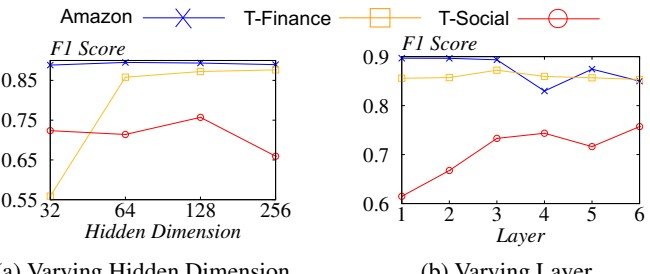

(a) Varying Hidden Dimension  (b) Varying Layer

Figure 5: Varying the Hidden Dimension and Layer.

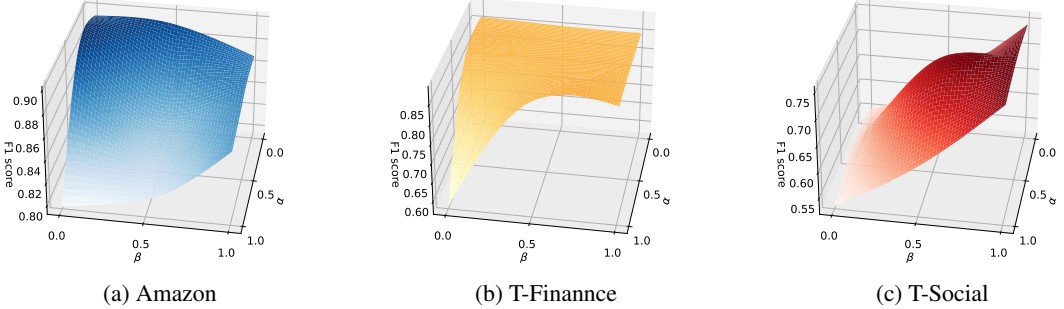

(a) Amazon  (b) T-Finannce  (c) T-Social

Figure 6: Varying $\alpha$ and $\beta$ on different datasets.

We provide the detailed algorithm in this Section. In Algorithm 1, we calculate $exp_{\boldsymbol{o}}^{\kappa}(\cdot)$ and $log_{\boldsymbol{o}}^{\kappa}(\cdot)$ based on the original point $\boldsymbol{o}$ of the space with curvature $\kappa$. Besides, to satisfy the range of $log_{\boldsymbol{o}}^{\kappa}(\cdot)$, we utilized Algorithm 2 to prune the node representations. Moreover, we construct Algorithm 3 by utilizing Algorithms 1 and 2. Specifically, we use the approximated distance to calculate the similarities between nodes and their neighbors, and then leverage them as the corresponding coefficients during the propagation process. Notice, for each layer $l$ during the propagation, we assign a different learnable $\kappa^l$ to capture comprehensive information from different spaces. To simplify our architecture, we only use three base models, $f_{\boldsymbol{\kappa}+}^L$, $f_{\boldsymbol{\kappa}-}^L$, and $f_{\boldsymbol{0}}^L$, for constructing SpaceGNN. This simplification can reduce the running time cost, and allow us to investigate the effectiveness of different spaces on different datasets easily through the corresponding hyperparameters. After obtaining probability matrix $\boldsymbol{Z}$ from Algorithm 4, we use the cross-entropy loss to update the framework.

# D EXPERIMENTAL SETTINGS

Table 4 provides a comprehensive list of our hyperparameters. We use grid search to train the model that yields the best F1 score on the validation set and report the corresponding test performance. Specifically, Learning Rate is searched from the set $\{0.001, 0.0001\}$, Hidden Dimension is chosen from the set $\{32, 64, 128, 256\}$, Layer ranges from 1 to 6, Dropout is obtained from the set $\{0, 0.05, 0.1\}$, Batch Size is fixed based on the size of the training set, and $\alpha$ and $\beta$ are from the set $\{0, 0.5, 1\}$, respectively. In the following Section E, we will further analyze the influence of Hidden Dimension, Layer, and $\alpha$ and $\beta$ on the F1 scores of different datasets.

Table 5: Ablation study.

| Datasets | Weibo | | Reddit | | Tolokers | | Amazon | | T-Finance | | YelpChi | | Questions | | DGraph-Fin | | T-Social | |
|---|---|---|---|---|---|---|---|---|---|---|---|---|---|---|---|---|---|---|
| Metrics | AUC | F1 | AUC | F1 | AUC | F1 | AUC | F1 | AUC | F1 | AUC | F1 | AUC | F1 | AUC | F1 | AUC | F1 |
| SpaceGNN | 0.9389 | 0.8541 | 0.6159 | 0.4915 | 0.7089 | 0.6012 | 0.9331 | 0.8935 | 0.9400 | 0.8723 | 0.6566 | 0.5719 | 0.6510 | 0.5336 | 0.6548 | 0.5017 | 0.9392 | 0.7571 |
| w/o LSP | 0.9279 | 0.8336 | 0.5964 | 0.4915 | 0.6964 | 0.5560 | 0.9135 | 0.8896 | 0.9384 | 0.8485 | 0.6274 | 0.5414 | 0.6445 | 0.5311 | 0.6465 | 0.5010 | 0.9042 | 0.7287 |
| w/o DAP | 0.9276 | 0.8442 | 0.5966 | 0.4915 | 0.6797 | 0.5717 | 0.9158 | 0.8814 | 0.9349 | 0.8553 | 0.6221 | 0.5621 | 0.6246 | 0.5300 | 0.6430 | 0.5001 | 0.9366 | 0.7353 |

Table 6: AUC and F1 scores (%) on 9 datasets with random split, compared with generalized models, where OOM represents out-of-memory.

| Datasets | Metrics | MLP | GCN | SAGE | GAT | GIN | HNN | HGCN | HYLA | SpaceGNN |
|---|---|---|---|---|---|---|---|---|---|---|
| Weibo | AUC | 0.2856 | 0.6223 | 0.6176 | 0.8077 | 0.4452 | 0.4844 | 0.8020 | **0.9351** | 0.8364 |
| | F1 | 0.5347 | 0.6215 | 0.4732 | 0.5127 | 0.5230 | 0.4727 | 0.4721 | 0.7008 | **0.7158** |
| Reddit | AUC | 0.5320 | 0.5865 | 0.5820 | 0.5421 | 0.5217 | 0.5266 | 0.5196 | 0.4734 | **0.5868** |
| | F1 | 0.4916 | 0.4916 | 0.4916 | 0.4916 | 0.4916 | 0.4916 | 0.4916 | 0.4916 | **0.4916** |
| Tolokers | AUC | 0.4632 | 0.5355 | 0.4119 | 0.6453 | 0.6030 | 0.5718 | 0.6495 | 0.4927 | **0.6952** |
| | F1 | 0.4375 | 0.5093 | 0.4423 | 0.5075 | 0.5621 | 0.5127 | 0.5522 | 0.5004 | **0.6026** |
| Amazon | AUC | 0.8550 | 0.7954 | 0.6162 | 0.5210 | 0.8254 | 0.7098 | 0.7468 | 0.7185 | **0.8722** |
| | F1 | 0.8261 | 0.6441 | 0.3725 | 0.3668 | 0.2256 | 0.4822 | 0.4822 | 0.5879 | **0.8641** |
| T-Finance | AUC | 0.9058 | 0.6796 | 0.6512 | 0.6327 | 0.7567 | 0.8761 | 0.0719 | 0.3917 | **0.9349** |
| | F1 | 0.7856 | 0.3068 | 0.4627 | 0.5293 | 0.7681 | **0.8204** | 0.4883 | 0.4919 | 0.8031 |
| YelpChi | AUC | 0.5064 | 0.4845 | 0.4975 | 0.5546 | 0.6082 | 0.3702 | 0.4745 | 0.5426 | **0.6191** |
| | F1 | 0.5064 | 0.4608 | 0.4980 | 0.5220 | 0.5332 | 0.4608 | 0.4608 | 0.4644 | **0.5475** |
| Questions | AUC | 0.5299 | 0.4684 | 0.5654 | 0.5554 | 0.5597 | 0.5177 | 0.5057 | 0.4049 | **0.5851** |
| | F1 | 0.4645 | 0.4924 | 0.4371 | 0.4923 | 0.4492 | **0.5089** | 0.5083 | 0.4924 | 0.4924 |
| DGraph-Fin | AUC | 0.4356 | 0.3900 | 0.5794 | 0.4103 | 0.4088 | 0.3282 | 0.3322 | OOM | **0.6515** |
| | F1 | 0.4531 | 0.4815 | 0.4994 | 0.4282 | 0.3787 | 0.4968 | 0.3312 | OOM | **0.5030** |
| T-Social | AUC | 0.5377 | 0.6183 | 0.6948 | 0.6958 | 0.5554 | 0.4694 | 0.4297 | OOM | **0.9019** |
| | F1 | 0.1373 | 0.2554 | 0.5421 | 0.5501 | 0.3733 | 0.4924 | 0.4923 | OOM | **0.7320** |

## E    PARAMETER ANALYSIS

In this Section, we investigate the impact of Hidden Dimension, Layer, and $\alpha$ and $\beta$ on three different datasets, and present their F1 scores.

Figure 5 reports the F1 score of SpaceGNN as we vary the Hidden Dimension from 32 to 256, and the Layer from 1 to 6. As we can observe, when we set the Hidden Dimension to 128, SpaceGNN achieves relatively satisfactory performances on these three datasets. When we vary the Layer, we find for different datasets, the optimal value can be different. Specifically, we set it to 3 for Amazon and T-Finance, and 6 for T-Social to get the best performance.

Figure 6 reports the F1 score of SpaceGNN as we vary $\alpha$ and $\beta$ from 0 to 1. These two hyperparameters represent the influence of diverse spaces on the datasets, so for different datasets, the optimal value will be distinct. Specifically, we set $\alpha$ to 0 for Amazon, 1 for T-Finance, and 0.5 for T-Social, and we set $\beta$ to 0 for Amazon, 1 for T-Finance and T-Social to get the satisfactory performance.

## F    ABLATION STUDY

To investigate the usefulness of the LSP and DAP components, we provide the ablation study of them on 9 datasets in Table 5. Specifically, we set the $\kappa$ as fixed values for different spaces during the w/o LSP experiment and set the $\hat{s}_i$ as $\mathbf{1}$ for each node $i$ during the w/o DAP experiment. As shown in Table 5, SpaceGNN consistently outperforms w/o LSP and w/o DAP by a large margin, which demonstrates the benefits of these two components.

## G    ADDITIONAL EXPERIMENTAL RESULTS

In addition to the experiments in Section 5, we further compare our SpaceGNN with baseline models on datasets with different sizes of training/validation/testing sets. Specifically, in experiments of Tables 6 and 7, we randomly divide each dataset into 10/10 for training/validation, and the rest of the nodes for testing, and in experiments of Tables 8 and 9, we randomly divide each dataset into 100/100 for training/validation, and the rest of the nodes for testing.

Table 7: AUC and F1 scores (%) on 9 datasets with random split, compared with specialized models, where TLE represents the experiment can not be conducted successfully within 72 hours.

| Datasets | Metrics | AMNet | BWGNN | GDN | SparseGAD | GHRN | GAGA | XGBGraph | CONSISGAD | SpaceGNN |
|---|---|---|---|---|---|---|---|---|---|---|
| Weibo | AUC | 0.4206 | 0.7557 | 0.7751 | 0.4558 | 0.6349 | 0.7597 | 0.5660 | 0.3972 | **0.8364** |
| | F1 | 0.5328 | 0.7106 | 0.1664 | 0.4724 | 0.6379 | 0.6574 | 0.5061 | 0.4447 | **0.7158** |
| Reddit | AUC | **0.6002** | 0.5815 | 0.4436 | 0.5263 | 0.5513 | 0.5068 | 0.5030 | 0.5536 | 0.5868 |
| | F1 | 0.4365 | 0.4617 | 0.4916 | 0.4721 | 0.4093 | 0.4916 | 0.4916 | 0.4514 | **0.4916** |
| Tolokers | AUC | 0.5627 | 0.5725 | 0.6159 | 0.4792 | 0.5688 | 0.6327 | 0.6083 | 0.5843 | **0.6952** |
| | F1 | 0.4451 | 0.5312 | 0.4665 | 0.4388 | 0.5449 | 0.4388 | 0.4989 | 0.5258 | **0.6026** |
| Amazon | AUC | 0.8356 | 0.7702 | 0.8335 | 0.7249 | 0.8028 | 0.7795 | 0.7666 | 0.8435 | **0.8722** |
| | F1 | 0.7144 | 0.5834 | 0.1685 | 0.4822 | 0.6777 | 0.6674 | 0.4822 | 0.8475 | **0.8641** |
| T-Finance | AUC | 0.8302 | 0.7318 | 0.5899 | 0.3650 | 0.7895 | 0.8157 | 0.8570 | 0.8503 | **0.9349** |
| | F1 | 0.5692 | 0.5025 | 0.5568 | 0.4883 | 0.5652 | 0.4894 | 0.7406 | **0.8316** | 0.8031 |
| YelpChi | AUC | 0.4738 | 0.5058 | 0.4893 | 0.5190 | 0.4231 | 0.4671 | 0.4927 | 0.5927 | **0.6191** |
| | F1 | 0.4875 | 0.4614 | 0.4977 | 0.4608 | 0.4608 | 0.4919 | 0.4608 | 0.5403 | **0.5475** |
| Questions | AUC | 0.4971 | 0.4125 | 0.5094 | 0.5185 | 0.5062 | 0.5361 | 0.5122 | 0.5492 | **0.5851** |
| | F1 | 0.4843 | 0.4924 | 0.4855 | 0.4988 | **0.5125** | 0.4944 | 0.4924 | 0.4935 | 0.4924 |
| DGraph-Fin | AUC | 0.3812 | 0.6343 | 0.3200 | 0.3346 | 0.3734 | TLE | 0.5009 | 0.6469 | **0.6515** |
| | F1 | 0.4128 | 0.4909 | 0.2641 | 0.4970 | 0.4871 | TLE | 0.4968 | 0.4224 | **0.5030** |
| T-Social | AUC | 0.4745 | 0.6408 | 0.5480 | 0.3317 | 0.6319 | TLE | 0.5066 | 0.8614 | **0.9019** |
| | F1 | 0.4810 | 0.4487 | 0.5124 | 0.4923 | 0.3435 | TLE | 0.4923 | 0.5890 | **0.7320** |

Table 8: AUC and F1 scores (%) on 9 datasets with random split, compared with generalized models, where OOM represents out-of-memory.

| Datasets | Metrics | MLP | GCN | SAGE | GAT | GIN | HNN | HGCN | HYLA | SpaceGNN |
|---|---|---|---|---|---|---|---|---|---|---|
| Weibo | AUC | 0.4438 | 0.9066 | 0.8157 | 0.8401 | 0.8541 | 0.7243 | 0.8545 | 0.9159 | **0.9521** |
| | F1 | 0.6538 | 0.8444 | 0.4746 | 0.7941 | 0.6561 | 0.6545 | 0.7620 | 0.4965 | **0.8481** |
| Reddit | AUC | 0.5907 | 0.5725 | 0.5767 | 0.6001 | 0.4793 | 0.5330 | 0.5315 | 0.4714 | **0.6232** |
| | F1 | 0.4916 | 0.4916 | 0.4916 | 0.4916 | 0.4916 | 0.4916 | 0.4916 | 0.4916 | **0.5228** |
| Tolokers | AUC | 0.7050 | 0.6952 | 0.7101 | 0.7139 | 0.7067 | 0.7063 | 0.7115 | 0.6402 | **0.7140** |
| | F1 | 0.5427 | 0.5978 | 0.5776 | 0.5844 | 0.5835 | 0.4711 | 0.5510 | 0.4864 | **0.6040** |
| Amazon | AUC | 0.8647 | 0.7936 | 0.7748 | 0.8808 | 0.9186 | 0.8635 | 0.7719 | 0.7188 | **0.9428** |
| | F1 | 0.7273 | 0.6167 | 0.6310 | 0.4354 | 0.7272 | 0.7711 | 0.5620 | 0.4822 | **0.9069** |
| T-Finance | AUC | 0.8960 | 0.8916 | 0.6722 | 0.8647 | 0.8087 | 0.8768 | 0.9329 | 0.3982 | **0.9486** |
| | F1 | 0.5737 | 0.7507 | 0.6042 | 0.8025 | 0.7680 | 0.8384 | 0.8753 | 0.4883 | **0.8789** |
| YelpChi | AUC | 0.7113 | 0.5160 | 0.5217 | 0.7249 | 0.7052 | 0.7119 | 0.5642 | 0.5508 | **0.7321** |
| | F1 | 0.6081 | 0.4608 | 0.4838 | 0.6256 | 0.6164 | 0.5919 | 0.4833 | 0.4608 | **0.6256** |
| Questions | AUC | 0.4707 | 0.6130 | 0.6000 | 0.5847 | 0.5083 | 0.5098 | 0.5081 | 0.4055 | **0.6476** |
| | F1 | 0.4961 | 0.4924 | 0.4999 | 0.5020 | 0.4819 | 0.4923 | 0.4924 | 0.4924 | **0.5386** |
| DGraph-Fin | AUC | 0.5752 | 0.6117 | 0.5487 | 0.6505 | 0.6408 | 0.3260 | 0.3298 | OOM | **0.6545** |
| | F1 | 0.4820 | 0.4769 | 0.4225 | 0.5000 | 0.5037 | 0.4968 | 0.3321 | OOM | **0.5097** |
| T-Social | AUC | 0.5896 | 0.7611 | 0.7268 | 0.6968 | 0.7248 | 0.5959 | 0.4245 | OOM | **0.9428** |
| | F1 | 0.4126 | 0.5666 | 0.5549 | 0.5770 | 0.5433 | 0.4936 | 0.4898 | OOM | **0.7828** |

In Tables 6 and 7, we can observe that our SpaceGNN can consistently surpass both generalized and specialized models on 9 datasets. In short, SpaceGNN outperforms the best rival 10.84% and 5.46% on average in terms of AUC and F1 scores, respectively.

Similarly, in Tables 8 and 9, it is easy to find out that SpaceGNN is able to beat both generalized and specialized models on 9 datasets. In summary, compared with the best rival, SpaceGNN takes a lead by 4.98% and 3.02% on average in terms of AUC and F1 scores, separately.

# H    ALTERNATIVE MODEL

The most common non-Euclidean GNN is based on either the Poincaré Ball model (Liu et al., 2019) or the Lorentz model (Nickel & Kiela, 2018). We discover that the Poincaré Ball model can be a special form of $\kappa$-stereographic model when setting the $\kappa$ to $-1$, which inspires us to investigate the general form of the Lorentz model. Following the definition of the $\kappa$-stereographic model, we generalize the Lorentz model as the $\kappa$-Lorentz model. Notice that we only provide a similar form to the $\kappa$-stereographic model, serving as the projection functions without considering the physical

Table 9: AUC and F1 scores (%) on 9 datasets with random split, compared with specialized models, where TLE represents the experiment can not be conducted successfully within 72 hours.

| Datasets | Metrics | AMNet | BWGNN | GDN | SparseGAD | GHRN | GAGA | XGBGraph | CONSISGAD | SpaceGNN |
|---|---|---|---|---|---|---|---|---|---|---|
| Weibo | AUC | 0.6902 | 0.8430 | 0.4353 | 0.8570 | 0.8286 | 0.8283 | 0.9496 | 0.7838 | **0.9521** |
| | F1 | 0.7156 | 0.7925 | 0.6680 | 0.6369 | 0.7918 | 0.7433 | 0.7431 | 0.7217 | **0.8481** |
| Reddit | AUC | 0.6011 | 0.5833 | 0.5840 | 0.4864 | 0.5823 | 0.4430 | 0.5518 | 0.5704 | **0.6232** |
| | F1 | 0.4916 | 0.4916 | 0.4916 | 0.4916 | 0.4916 | 0.4916 | 0.4909 | 0.4916 | **0.5228** |
| Tolokers | AUC | 0.6939 | 0.7100 | **0.7325** | 0.6879 | 0.7197 | 0.4817 | 0.6804 | 0.7088 | 0.7140 |
| | F1 | 0.5910 | 0.5943 | 0.5834 | 0.4711 | 0.5871 | 0.2794 | 0.5954 | 0.5932 | **0.6040** |
| Amazon | AUC | 0.8812 | 0.8742 | 0.8939 | 0.7263 | 0.8843 | 0.7476 | 0.9124 | 0.9325 | **0.9428** |
| | F1 | 0.8768 | 0.8893 | 0.8732 | 0.5939 | 0.8286 | 0.7224 | 0.8607 | 0.8990 | **0.9069** |
| T-Finance | AUC | 0.7774 | 0.8907 | 0.7863 | 0.9247 | 0.8982 | 0.8387 | 0.9407 | 0.9359 | **0.9486** |
| | F1 | 0.7528 | 0.7726 | 0.7779 | 0.4883 | 0.7805 | 0.5482 | **0.8831** | 0.8722 | 0.8789 |
| YelpChi | AUC | 0.7201 | 0.7022 | 0.7165 | 0.5504 | 0.6974 | 0.5107 | 0.7239 | 0.7152 | **0.7321** |
| | F1 | 0.6202 | 0.6119 | 0.6161 | 0.4608 | 0.5575 | 0.4608 | 0.6232 | 0.6187 | **0.6256** |
| Questions | AUC | 0.5959 | 0.5939 | 0.4870 | 0.5080 | 0.5931 | 0.5088 | 0.5217 | 0.5652 | **0.6476** |
| | F1 | 0.5145 | 0.4992 | 0.4936 | 0.4955 | 0.4993 | 0.4924 | 0.4926 | 0.5174 | **0.5386** |
| DGraph-Fin | AUC | 0.5392 | 0.5887 | 0.5407 | 0.3418 | 0.6012 | TLE | 0.5080 | 0.5108 | **0.6545** |
| | F1 | 0.5043 | 0.5073 | 0.4968 | 0.4487 | 0.4973 | TLE | 0.4974 | 0.5058 | **0.5097** |
| T-Social | AUC | 0.5114 | 0.7544 | 0.4846 | 0.7199 | 0.6353 | TLE | 0.7381 | 0.9192 | **0.9428** |
| | F1 | 0.4653 | 0.5731 | 0.4697 | 0.4924 | 0.4923 | TLE | 0.5547 | 0.7170 | **0.7828** |

Table 10: Alternative framework.

| Datasets | Weibo | | Reddit | | Tolokers | | Amazon | | T-Finance | | YelpChi | | Questions | | DGraph-Fin | | T-Social | |
|---|---|---|---|---|---|---|---|---|---|---|---|---|---|---|---|---|---|---|
| Metrics | AUC | F1 | AUC | F1 | AUC | F1 | AUC | F1 | AUC | F1 | AUC | F1 | AUC | F1 | AUC | F1 | AUC | F1 |
| SpaceGNN | 0.9389 | 0.8541 | 0.6159 | 0.4915 | 0.7089 | 0.6012 | 0.9331 | 0.8935 | 0.9400 | 0.8723 | 0.6566 | 0.5719 | 0.6510 | 0.5336 | 0.6548 | 0.5017 | 0.9392 | 0.7571 |
| SpaceGNN-L | 0.9389 | 0.8550 | 0.5900 | 0.4915 | 0.7035 | 0.5818 | 0.9155 | 0.8911 | 0.9251 | 0.8728 | 0.6610 | 0.5577 | 0.6403 | 0.5356 | 0.6407 | 0.5008 | 0.9392 | 0.7469 |

meaning. The $exp_o^\kappa(\cdot)$ and $log_o^\kappa(\cdot)$ for $\boldsymbol{x} \in \mathbb{R}^d$ are defined as follows:

$$exp_{\boldsymbol{x}'}^\kappa(\boldsymbol{x}) = cos_\kappa(||\boldsymbol{x}||_L)\boldsymbol{x}' + sin_\kappa(||\boldsymbol{x}||_L)\frac{\boldsymbol{x}}{||\boldsymbol{x}||_L}$$

$$log_{\boldsymbol{x}'}^\kappa(\boldsymbol{x}) = d_\kappa(\boldsymbol{x}, \boldsymbol{x}')\frac{\boldsymbol{x} + \frac{1}{\kappa}\langle \boldsymbol{x}, \boldsymbol{x}'\rangle_L \boldsymbol{x}'}{||\boldsymbol{x} + \frac{1}{\kappa}\langle \boldsymbol{x}, \boldsymbol{x}'\rangle_L \boldsymbol{x}'||_L}$$

where $\langle \boldsymbol{x}, \boldsymbol{x}'\rangle_L = -x_0 x_0' + x_1 x_1' + ... + x_d x_d'$, $||\boldsymbol{x}||_L = \sqrt{\langle \boldsymbol{x}, \boldsymbol{x}'\rangle_L}$, $d_\kappa(\boldsymbol{x}, \boldsymbol{x}') = cos_\kappa^{-1}(-\langle \boldsymbol{x}, \boldsymbol{x}'\rangle_L)$, and $cos_\kappa$ and $sin_\kappa$ are defined as:

$$cos_\kappa(\boldsymbol{x}) = \begin{cases} \frac{1}{\sqrt{-\kappa}}\cosh(\sqrt{-\kappa}\boldsymbol{x}), & \kappa < 0, \\ \boldsymbol{x}, & \kappa = 0, \\ \frac{1}{\sqrt{\kappa}}\cos(\sqrt{\kappa}\boldsymbol{x}), & \kappa > 0. \end{cases}$$

$$sin_\kappa(\boldsymbol{x}) = \begin{cases} \frac{1}{\sqrt{-\kappa}}\sinh(\sqrt{-\kappa}\boldsymbol{x}), & \kappa < 0, \\ \boldsymbol{x}, & \kappa = 0, \\ \frac{1}{\sqrt{\kappa}}\sin(\sqrt{\kappa}\boldsymbol{x}), & \kappa > 0. \end{cases}$$

We replace the corresponding functions in our SpaceGNN framework to get SpaceGNN-L. As shown in Table 10, SpaceGNN and SpaceGNN-L can have similar performance in terms of all the 9 datasets, which shows SpaceGNN-L can also outperform other baselines. These results demonstrate that our framework can be generalized to other base models.

# I   LEARNED $\kappa$

In this section, we report the learned $\boldsymbol{\kappa}$ of the experiments in Tables 1 and 2. Notice that, for simplicity, we only include 1 Euclidean GNN, 1 Hyperbolic GNN, and 1 Spherical GNN in our framework. Recap from Section 4, in our final architecture, we utilize a hyperparameter $L$ to control the number of layers of all three GNNs, and the number of entries in $\boldsymbol{\kappa}$ for each GNN is the same as the number of layers of it. Specifically, if $L$ is set to be 6, then there will be 6 entries in $\boldsymbol{\kappa}^0$ for Euclidean GNN, 6 entries in $\boldsymbol{\kappa}^-$ for Hyperbolic GNN, and 6 entries in $\boldsymbol{\kappa}^+$ for Spherical GNN. For $\boldsymbol{\kappa}^0$, we want the GNN to stay in the Euclidean space, so we set each entry in it to 0. For $\boldsymbol{\kappa}^-$ and $\boldsymbol{\kappa}^+$, we want the Hyperbolic GNN and Spherical GNN to search for the optimal curvatures for different

Table 11: Learned $\kappa$

| Datasets | $\kappa_1^-$ | $\kappa_2^-$ | $\kappa_3^-$ | $\kappa_4^-$ | $\kappa_5^-$ | $\kappa_6^-$ | $\kappa_1^+$ | $\kappa_2^+$ | $\kappa_3^+$ | $\kappa_4^+$ | $\kappa_5^+$ | $\kappa_6^+$ |
|---|---|---|---|---|---|---|---|---|---|---|---|---|
| Weibo | -0.1272 | -0.0766 | -0.0795 | -0.1176 | -0.1239 | -0.1103 | 0.0989 | 0.1233 | 0.0936 | 0.0961 | 0.0747 | 0.1116 |
| Reddit | -0.0898 | -0.1012 | -0.1028 | -0.1048 | -0.1106 | - | 0.0839 | 0.0990 | 0.1223 | 0.0939 | 0.0963 | - |
| Tolokers | -0.1046 | - | - | - | - | - | 0.1259 | - | - | - | - | - |
| Amazon | -0.0873 | -0.1095 | -0.1054 | - | - | - | 0.0768 | 0.0685 | 0.0864 | - | - | - |
| T-Finance | -0.1347 | -0.0701 | -0.0738 | - | - | - | 0.0744 | 0.0652 | 0.0850 | - | - | - |
| YelpChi | -0.1014 | - | - | - | - | - | 0.1369 | - | - | - | - | - |
| Questions | -0.0999 | -0.1013 | -0.0992 | -0.1004 | -0.0983 | -0.1008 | 0.1006 | 0.0998 | 0.0998 | 0.1006 | 0.1001 | 0.0998 |
| DGraph-Fin | -0.1262 | -0.0774 | -0.0803 | -0.1169 | - | - | 0.0784 | 0.0906 | 0.0994 | 0.1130 | - | - |
| T-Social | -0.1440 | -0.0621 | -0.0669 | -0.1284 | -0.1386 | -0.1167 | 0.0992 | 0.1181 | 0.0950 | 0.0970 | 0.0804 | 0.1090 |

datasets, so we set these two as learnable curvatures. Notice that, according to Table 4, the optimal $L$ varies by datasets, so the number of entries in learned $\kappa^-$ and $\kappa^+$ will also be different, as shown in Table 11, where " $-$ " represents no such entry in the vector.

As we can see from Table 11, the learned values stay close to 0 after the learning process, which is aligned with the analysis of Section 4.2. As shown in Figure 2, the largest $ER_\kappa$ will be obtained around 0, which further demonstrates our findings are effective for graph anomaly detection tasks.

## J    TIME COMPLEXITY ANALYSIS

As shown in Section 4.4, our framework is composed of 1 Euclidean GNN, $H$ Hyperbolic GNN, and $S$ Spherical GNN. The differences between these GNNs are the projection functions and the Distance Aware Propagation (DAP) component, but the time complexity of them is the same for different GNNs. Hence, We only need to analyze one of the GNNs.

Our analysis of the GNN time complexity is primarily based on the Algorithm 3 in Appendix C, which illustrates the base architecture of each GNN. For simplicity, we focus on a single layer in the base architecture (Lines 3-7).

First, in Line 3, we apply a two-layer MLP with time complexity of $O(|V|dd_1 + |V|d_1d_2)$ followed by a projection function with time complexity of $O(|V|d_2)$, where $|V|$ is the total number of nodes in the graph, $d$ is the dimension of the node feature, and $d_1, d_2$ are the output dimension of the two MLPs, respectively. Thus, the total time complexity of Line 3 is $O(|V|dd_1 + |V|d_1d_2)$.

Then, in Line 5, we have to calculate the $\hat{s}_i^{\kappa^l}$ for each node $i$. Specifically, for each edge connected to node $i$, we have a time complexity of $O(d_2)$ to get the corresponding coefficient. Thus, the total time complexity for all nodes in Line 5 would be $O(|E|d_2)$, where $|E|$ is the total number of edges in this graph.

Afterward, in Line 6, for each edge between nodes $i$ and $j$, we need to calculate the $\omega_{ij}^{\kappa^l}$ with time complexity of $O(d_2^2)$, where the input dimension of the MLP is $2d_2$ and the output dimension of it is $d_2$, so the total complexity for all edges in Line 6 would be $O(|E|d_2^2)$.

Next, in Line 7, we also have to propagate the node embeddings for each edge in the graph, so the total time complexity of Line 7 is $O(|E|d_2)$.

Finally, we combine the results before to get the time complexity of a single layer in the base architecture, i.e., $O(|V|dd_1 + |V|d_1d_2 + |E|d_2^2)$.

According to the time analysis of GAT (Velickovic et al., 2018), one of the most popular architectures in the area of graph learning, the time complexity of a single GAT attention head computing $F_0$ features can be expressed as $O(|V|FF_0 + |E|F_0)$, where $F$ is the number of input features, and $|V|$ and $|E|$ are the numbers of nodes and edges in the graph, respectively.

Hence, each layer of our proposed GNN has a similar time complexity to GAT by choosing the proper hyperparameters $d_1$ and $d_2$ in our architecture. In the experiments, we find that combining 1 Euclidean GNN, 1 Hyperbolic GNN, and 1 Spherical GNN in our framework is enough to achieve superior performance over all the other baselines, so the increase of time complexity by the Multiple Space Ensemble component will not be the limitation of our models in real applications.

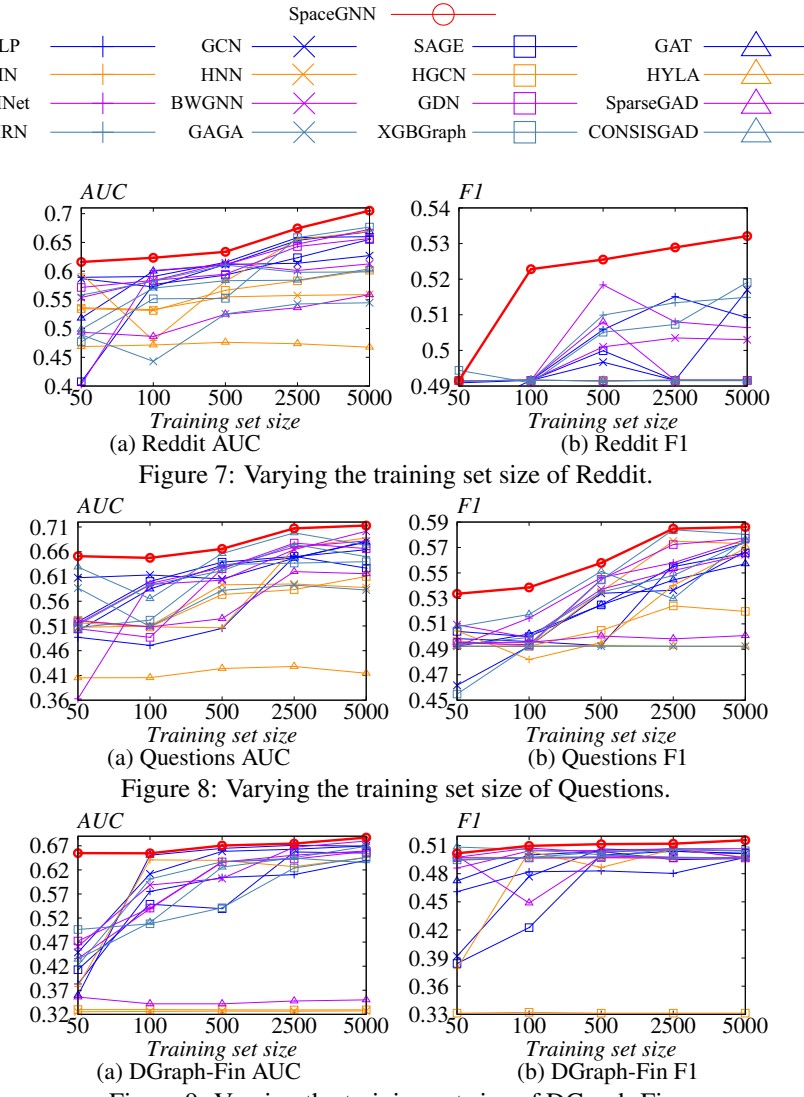

Figure 7: Varying the training set size of Reddit.

Figure 8: Varying the training set size of Questions.

Figure 9: Varying the training set size of DGraph-Fin.

## K    PERFORMANCE WITH MORE TRAINING DATA

To further demonstrate the superior ability of our proposed framework, we provide the performance on Reddit, Questions, and DGraph-Fin varying by the size of the training set, as shown in Figures 7, 8, and 9. Note that, since HYLA and GAGA can not successfully run on DGraph-Fin, we only report the performance of the other 14 baselines and our proposed SpaceGNN in Figure 9.

As we can see, the red lines, which represent the performance of our SpaceGNN, are always on the top of the figures, which demonstrates that with more training data, our SpaceGNN can still outperform all the other baselines consistently in terms of both AUC and F1. In summary, such experiments further make our SpaceGNN a more general and practical algorithm.

## L    PERFORMANCE ON GADBENCH (TANG ET AL., 2023) SEMI-SUPERVISED SETTING

For a fair comparison, we also provide AUC, AUPRC, and Rec@K scores on 9 datasets with data split of the semi-supervised setting in GADBench (Tang et al., 2023). Specifically, in this setting, we use 20 positive labels (anomalous nodes) and 80 negative labels (normal nodes) for both the training set and the validation set in each dataset, separately. Note that, for the baselines in GADBench,

Table 12: AUC, AUPRC, and Rec@K scores (%) on 9 datasets with data split of the semi-supervised setting in GADBench (Tang et al., 2023), compared with generalized models, where OOM represents out-of-memory.

| Datasets | Metrics | MLP | GCN | SAGE | GAT | GIN | HNN | HGCN | HYLA | SpaceGNN |
|---|---|---|---|---|---|---|---|---|---|---|
| Weibo | AUC | 0.666 | 0.935 | 0.818 | 0.864 | 0.838 | 0.747 | 0.942 | 0.960 | **0.964** |
| | AUPRC | 0.562 | 0.860 | 0.585 | 0.733 | 0.676 | 0.312 | 0.808 | 0.727 | **0.864** |
| | Rec@K | 0.532 | 0.792 | 0.634 | 0.702 | 0.665 | 0.371 | 0.757 | 0.736 | **0.795** |
| Reddit | AUC | 0.591 | 0.569 | 0.603 | 0.605 | 0.600 | 0.619 | 0.625 | 0.523 | **0.637** |
| | AUPRC | 0.044 | 0.042 | 0.045 | 0.047 | 0.043 | 0.045 | 0.045 | 0.038 | **0.050** |
| | Rec@K | 0.065 | 0.062 | 0.058 | 0.065 | 0.048 | 0.055 | 0.052 | 0.064 | **0.077** |
| Tolokers | AUC | 0.681 | 0.642 | 0.676 | 0.681 | 0.668 | 0.690 | 0.699 | 0.618 | **0.715** |
| | AUPRC | 0.333 | 0.330 | 0.340 | 0.330 | 0.318 | 0.327 | 0.335 | 0.290 | **0.362** |
| | Rec@K | 0.355 | 0.334 | 0.352 | 0.351 | 0.336 | 0.346 | 0.351 | 0.311 | **0.366** |
| Amazon | AUC | 0.922 | 0.820 | 0.814 | 0.924 | 0.916 | 0.861 | 0.792 | 0.717 | **0.947** |
| | AUPRC | **0.830** | 0.328 | 0.425 | 0.816 | 0.754 | 0.785 | 0.306 | 0.168 | 0.812 |
| | Rec@K | **0.793** | 0.369 | 0.480 | 0.771 | 0.704 | 0.776 | 0.356 | 0.236 | 0.782 |
| T-Finance | AUC | 0.899 | 0.883 | 0.689 | 0.850 | 0.845 | 0.880 | 0.933 | 0.615 | **0.949** |
| | AUPRC | 0.534 | 0.605 | 0.117 | 0.289 | 0.448 | 0.677 | 0.799 | 0.063 | **0.849** |
| | Rec@K | 0.599 | 0.606 | 0.185 | 0.362 | 0.544 | 0.638 | 0.760 | 0.080 | **0.796** |
| YelpChi | AUC | 0.647 | 0.512 | 0.589 | 0.656 | 0.629 | 0.662 | 0.480 | 0.551 | **0.726** |
| | AUPRC | 0.236 | 0.164 | 0.209 | 0.250 | 0.237 | 0.263 | 0.141 | 0.174 | **0.331** |
| | Rec@K | 0.265 | 0.169 | 0.229 | 0.281 | 0.265 | 0.296 | 0.149 | 0.200 | **0.366** |
| Questions | AUC | 0.612 | 0.600 | 0.612 | 0.623 | 0.622 | 0.601 | 0.575 | 0.619 | **0.650** |
| | AUPRC | 0.077 | 0.061 | 0.055 | 0.073 | 0.067 | 0.047 | 0.039 | 0.057 | **0.097** |
| | Rec@K | 0.120 | 0.098 | 0.088 | 0.109 | 0.103 | 0.051 | 0.030 | 0.106 | **0.145** |
| DGraph-Fin | AUC | **0.691** | 0.662 | 0.648 | 0.672 | 0.657 | 0.644 | 0.638 | OOM | 0.678 |
| | AUPRC | 0.023 | 0.023 | 0.020 | 0.022 | 0.020 | 0.022 | 0.023 | OOM | **0.025** |
| | Rec@K | 0.034 | 0.036 | 0.025 | 0.031 | 0.021 | 0.013 | 0.027 | OOM | **0.040** |
| T-Social | AUC | 0.591 | 0.716 | 0.720 | 0.754 | 0.704 | 0.473 | 0.435 | OOM | **0.947** |
| | AUPRC | 0.039 | 0.084 | 0.078 | 0.092 | 0.062 | 0.027 | 0.024 | OOM | **0.642** |
| | Rec@K | 0.032 | 0.102 | 0.095 | 0.116 | 0.053 | 0.009 | 0.001 | OOM | **0.667** |

Table 13: AUC, AUPRC, and Rec@K scores (%) on 9 datasets with data split of the semi-supervised setting in GADBench (Tang et al., 2023), compared with specialized models, where TLE represents the experiment can not be conducted successfully within 72 hours.

| Datasets | Metrics | AMNet | BWGNN | GDN | SparseGAD | GHRN | GAGA | XGBGraph | CONSISGAD | SpaceGNN |
|---|---|---|---|---|---|---|---|---|---|---|
| Weibo | AUC | 0.824 | 0.936 | 0.682 | 0.897 | 0.916 | 0.732 | **0.964** | 0.873 | **0.964** |
| | AUPRC | 0.671 | 0.806 | 0.582 | 0.696 | 0.770 | 0.376 | 0.759 | 0.654 | **0.864** |
| | Rec@K | 0.621 | 0.751 | 0.560 | 0.678 | 0.724 | 0.324 | 0.689 | 0.583 | **0.795** |
| Reddit | AUC | 0.629 | 0.577 | 0.596 | 0.634 | 0.575 | 0.501 | 0.592 | 0.629 | **0.637** |
| | AUPRC | 0.049 | 0.042 | 0.043 | 0.047 | 0.042 | 0.032 | 0.041 | 0.046 | **0.050** |
| | Rec@K | 0.068 | 0.060 | 0.052 | 0.074 | 0.063 | 0.019 | 0.049 | 0.061 | **0.077** |
| Tolokers | AUC | 0.617 | 0.685 | 0.713 | 0.673 | 0.690 | 0.636 | 0.675 | 0.709 | **0.715** |
| | AUPRC | 0.286 | 0.353 | 0.353 | 0.318 | 0.359 | 0.293 | 0.341 | 0.337 | **0.362** |
| | Rec@K | 0.305 | 0.355 | 0.363 | 0.346 | 0.361 | 0.318 | **0.366** | 0.364 | **0.366** |
| Amazon | AUC | 0.928 | 0.918 | 0.868 | 0.935 | 0.909 | 0.504 | **0.947** | 0.933 | **0.947** |
| | AUPRC | 0.824 | 0.817 | 0.691 | 0.800 | 0.807 | 0.148 | **0.844** | 0.792 | 0.812 |
| | Rec@K | 0.778 | 0.777 | 0.652 | **0.788** | 0.777 | 0.143 | 0.782 | 0.775 | 0.782 |
| T-Finance | AUC | 0.926 | 0.921 | 0.900 | 0.944 | 0.926 | 0.725 | 0.948 | 0.932 | **0.949** |
| | AUPRC | 0.602 | 0.609 | 0.671 | 0.835 | 0.634 | 0.252 | 0.783 | 0.815 | **0.849** |
| | Rec@K | 0.657 | 0.649 | 0.656 | 0.794 | 0.677 | 0.400 | 0.724 | 0.758 | **0.796** |
| YelpChi | AUC | 0.648 | 0.643 | 0.670 | 0.639 | 0.645 | 0.549 | 0.640 | 0.715 | **0.726** |
| | AUPRC | 0.239 | 0.237 | 0.244 | 0.213 | 0.238 | 0.173 | 0.248 | 0.330 | **0.331** |
| | Rec@K | 0.266 | 0.264 | 0.278 | 0.222 | 0.269 | 0.187 | 0.268 | 0.358 | **0.366** |
| Questions | AUC | 0.636 | 0.602 | 0.609 | 0.574 | 0.605 | 0.513 | 0.614 | 0.649 | **0.650** |
| | AUPRC | 0.074 | 0.065 | 0.070 | 0.036 | 0.065 | 0.039 | 0.077 | 0.085 | **0.097** |
| | Rec@K | 0.127 | 0.109 | 0.097 | 0.032 | 0.111 | 0.072 | 0.106 | 0.092 | **0.145** |
| DGraph-Fin | AUC | 0.671 | 0.655 | 0.660 | 0.674 | 0.671 | TLE | 0.624 | 0.635 | **0.678** |
| | AUPRC | 0.022 | 0.021 | 0.022 | 0.023 | 0.023 | TLE | 0.019 | 0.017 | **0.025** |
| | Rec@K | 0.026 | 0.031 | 0.032 | 0.022 | 0.034 | TLE | 0.025 | 0.011 | **0.040** |
| T-Social | AUC | 0.537 | 0.775 | 0.716 | 0.766 | 0.787 | TLE | 0.852 | 0.940 | **0.947** |
| | AUPRC | 0.031 | 0.159 | 0.104 | 0.256 | 0.162 | TLE | 0.406 | 0.484 | **0.642** |
| | Rec@K | 0.016 | 0.243 | 0.199 | 0.362 | 0.246 | TLE | 0.430 | 0.535 | **0.667** |

we use the reported performance in it, and for baselines not in GADBench, we obtain the source code of all competitors from GitHub and execute these models using the default parameter settings suggested by their authors. The hyperparameters of SpaceGNN are set based on the same setting in GADBench, i.e., random search.

As we can see from Tables 12 and 13, our proposed model can still outperform all the baselines on almost all the datasets consistently using AUC, AUPRC, and Rec@K scores as metrics, which demonstrates the effectiveness of our SpaceGNN.

