# OpenReview forum: "SpaceGNN: Multi-Space Graph Neural Network for Node Anomaly Detection with Extremely Limited Labels"
_ICLR.cc/2025/Conference — ICLR 2025 Poster_

### Official Review · Reviewer_Snna · 2024-10-30

**Soundness:** 3
**Presentation:** 3
**Contribution:** 3
**Rating:** 6
**Confidence:** 4

**Summary:**

This paper studies the problem of Node Anomaly Detection when the training data is as small as 50. The authors design a Learnable Space Projection function that effectively encodes nodes into suitable spaces, and a Multiple Space Ensemble module to extracts comprehensive information for NAD under conditions of extremely limited supervision. The most surprising result is that it is more beneficial than data augmentation techniques.

**Strengths:**

1. The method requires very limited training samples, which is a promising property in anomaly detection where anomaly training data is limited.

2. The paper is well-organized, with each section (subsection) clearly stating the intuition along with a solid derivation.

3. Baselines and datasets are sufficient. SpaceGNN consistently outperforms 8 baselines on 9 datasets, which demonstrates the effectiveness of the paper.

**Weaknesses:**

1. SpaceGNN has demonstrated its superior performance under low-resource conditions. How about the performance when more training data are available? This experiment may make SpaceGNN a more general and practical algorithm.

**Questions:**

How about the performance when more training data are available?

---

> ### Author Response · Authors · 2024-11-19
> **Author Response for Reviewer Snna**
>
> We appreciate your comprehensive and constructive review. Your crucial comments on the statistical differences between anomalous graphs and normal graphs are exceedingly helpful for us to improve our manuscript. Our point-to-point responses to your comments are given below.
>
> ---
>
> **W1**: SpaceGNN has demonstrated its superior performance under low-resource conditions. How about the performance when more training data are available? This experiment may make SpaceGNN a more general and practical algorithm.
>
> **RW1**:  Thank you for pointing that out, We have updated our manuscript and included performance figures for various datasets across different training set sizes in Appendix K. Through these additional experiments, we can observe that as the size of the training data increases, our proposed SpaceGNN still outperforms all baseline models. For detailed discussions, please kindly refer to our revised manuscript.
>
> ---
>
> **Q1**: How about the performance when more training data are available?
>
> **RQ1**: Please kindly refer to **RW1**.
>
> ---
>
> We sincerely appreciate your time, and we are glad to answer any additional questions you may have.

---

> > ### Comment · Reviewer_Snna · 2024-11-23
> >
> > Dear author,
> >
> > Thanks for the new experiment. It demonstrates that SpaceGNN could be applied to more scenarios. I'd like to maintain my positive score.

---

> > > ### Author Response · Authors · 2024-12-03
> > >
> > > Dear Reviewer,
> > >
> > > Thank you very much for the time and effort you have put into reviewing our paper and providing insightful comments. Moreover, we are also delighted to know that you are satisfied with our responses and confirm our contribution to the graph anomaly detection area. We deeply appreciate your support for our work.
> > >
> > > Warm regards,
> > >
> > > The Authors of Submission6161

---

> ### Author Response · Authors · 2024-11-23
> **Author Response for Reviewer Snna**
>
> Dear Reviewer,
>
> We sincerely hope that we have addressed all your concerns, and we are glad to answer any additional questions you may have.
>
> Warm regards,
>
> The Authors of Submission6161

---

### Official Review · Reviewer_W8L4 · 2024-11-03

**Soundness:** 3
**Presentation:** 2
**Contribution:** 2
**Rating:** 5
**Confidence:** 2

**Summary:**

This paper studies the node classification problem by ensembling embeddings from multiple spaces.

**Strengths:**

S1. This paper includes theoretical analyses to show the rationale of the proposed method.

S2. This paper includes many datasets and comprehensive baseline methods.

**Weaknesses:**

W1. I think this paper has two important focuses: (1) the anomaly detection task and (2) extremely limited samples. However, after reading the whole methodology section, my personal impression is that the proposed method is designed for a regular binary classification task. I suggest the authors emphasize more on which designs are specialized for few-shot samples and what makes this proposed method specialized for an anomaly detection problem, compared to a regular binary classification task.

W2. This paper includes 4 theorems in total, but I found that theorems 3 and 4 are very general theorems, regarding the general ensemble methods for cross-entopy losses, which is irrelevant to the node anomaly detection task. Also, to be frank, I think they are a bit trivial because they are very straightforward and might not be appropriate to be termed as "theorems" (maybe propositions).

W3. Another concern of this paper is its writing. I tried to understand the whole model architecture through the equations in Section 4 but found some notations that were missing or had not been introduced appropriately. Also, it looks like most critical equations are not numbered. I suggest numbering them so they can be quoted when two equations are closely working. I will name a few:

W3.1 I found the distance is defined in line 236, but it looks like it is not being used in Eqs between lines 281 and 286.

W3.2 Eq. 405 defines the ensemble of models, which is the sum of several functions, but it is unclear how this "addition" operation works for several functions. Do you mean the output of every function from line 286 is added together?

**Questions:**

Please check the weaknesses I mentioned.

---

> ### Author Response · Authors · 2024-11-19
> **Author Response for Reviewer W8L4 (1/2)**
>
> We appreciate your comprehensive and constructive review. Your crucial comments on experiments are exceedingly helpful for us to improve our manuscript. Our point-to-point responses to your comments are given below.
>
> ---
>
> **W1**: I think this paper has two important focuses: (1) the anomaly detection task and (2) extremely limited samples. However, after reading the whole methodology section, my personal impression is that the proposed method is designed for a regular binary classification task. I suggest the authors emphasize more on which designs are specialized for few-shot samples and what makes this proposed method specialized for an anomaly detection problem, compared to a regular binary classification task.
>
> **RW1**: Thanks for pointing that out. However, in our manuscript, we have provided detailed explanations of why our proposed method is specifically designed for graph anomaly detection rather than a simple regular binary classification task.
>
> Firstly, in the real applications of regular binary classification tasks, the complexity of graph structures is typically less pronounced when compared to the intricate structures found in real-world graph anomaly detection tasks, e.g., tree structures in malicious review detection or circle structures in money laundry detection. In Section 1, we have introduced that these structures represent a critical challenge in addressing real graph anomaly tasks. In Section 4, we have also provided both theoretical analyses and empirical evidence to show that it is better to project such complex structures into different spaces using our proposed components.
>
> Secondly, regular binary classification tasks usually do not suffer from the limited supervision problem in the real deployment. However, it is a ubiquitous problem within the area of graph anomaly detection tasks. Building on such a fact, we apply the ensemble technique to collect comprehensive information from different independent spaces to boost the performance. Our theoretical analysis elucidates the efficacy of the ensemble method for tackling such challenges. Thus, From both theoretical and empirical perspectives, our framework is highly correlated to graph anomaly detection tasks.
>
> ---
>
> **W2**: This paper includes 4 theorems in total, but I found that theorems 3 and 4 are very general theorems, regarding the general ensemble methods for cross-entopy losses, which is irrelevant to the node anomaly detection task. Also, to be frank, I think they are a bit trivial because they are very straightforward and might not be appropriate to be termed as "theorems" (maybe propositions).
>
> **RW2**: Thanks for pointing that out. The inclusion of these two theoretical analyses serves as a crucial justification, aligning with the subsequent experiments detailed in Section 5. These analyses are not trivial since the most important prerequisite of these two analyses is the independence of different models. Notably, all the previous works in this domain, as discussed in Sections 1 and 2, are designed in a single space, and thus they cannot leverage ensemble methods as our proposed framework. With three independent spaces, i.e., Hyperbolic, Euclidean, and Spherical spaces, our model is designed to employ an ensemble architecture, specifically tailored to address the challenges posed by limited supervision in practical graph anomaly detection tasks. Therefore, it is necessary to present theoretical analyses of the ensemble method, given its significant relevance to our approach. However, in response to your constructive feedback, we have implemented the suggested revisions by transitioning Theorems 3 and 4 to Propositions 1 and 2 in our manuscript. Please kindly refer to our revised manuscript.

---

> ### Author Response · Authors · 2024-11-19
> **Author Response for Reviewer W8L4 (2/2)**
>
> ---
>
> **W3**: Another concern of this paper is its writing. I tried to understand the whole model architecture through the equations in Section 4 but found some notations that were missing or had not been introduced appropriately. Also, it looks like most critical equations are not numbered. I suggest numbering them so they can be quoted when two equations are closely working.
>
> **RW3**: Thanks for pointing that out. We have revised our manuscript and added numbers to the critical equations. Please kindly refer to the updated version of our manuscript.
>
> ---
>
> **W3.1**: I found the distance is defined in line 236, but it looks like it is not being used in Eqs between lines 281 and 286.
>
> **RW3.1**: We have indeed incorporated the distance concept into our framework as a pivotal aspect of our contributions. The definition of distance in Line 236 serves to elucidate a crucial observation we have made, i.e., the Expansion Rate (defined based on the distance in Definition 1) demonstrates that for different data, it is better to project the node features into different spaces to collect comprehensive information, thereby enhancing model performance. In the context of Lines 281-286, where we outline our base GNN architecture in a general form, the coefficient $\omega_{ij}^{\kappa^l}$ in Line 341 is defined based on the distance in Line 236. This linkage ensures that our framework can capture the underlying properties of distances within the neighborhood of all the nodes.
>
> ---
>
> **W3.2**: Eq. 405 defines the ensemble of models, which is the sum of several functions, but it is unclear how this "addition" operation works for several functions. Do you mean the output of every function from line 286 is added together?
>
> **RW3.2**: The “addition” operation is similar to the soft-voting algorithm. In the design of ensemble models, the soft-voting algorithm is one of the most prevalent techniques. Initially, each classifier assigns probabilities to individual classes, and their averages are computed. The ensemble prediction is then determined by the class with the highest cumulative probability. Notably, while the original soft-voting algorithm computes the arithmetic average of all classifiers, we implement a weighted average soft-voting to better adjust the weights of different classifiers. For example, consider a scenario with three classifiers trained to classify nodes as anomalies or normal. If the first classifier predicts anomaly with a probability of 0.7 and normal with 0.3, the second classifier predicts anomaly with a probability of 0.4 and normal with 0.6, and the third classifier predicts anomaly with a probability of 0.9 and normal with 0.1. Assigning weights of 0.1, 0.5, and 0.4 to the first, second, and third classifiers, respectively, the ensemble would predict that the node is an anomaly based on weighted probabilities: $0.7 * 0.1 + 0.4 * 0.5 + 0.9 * 0.4 = 0.63$ for anomaly and $0.3 * 0.1 + 0.6 * 0.5 + 0.1 * 0.4 = 0.37$ for normal class probabilities.
>
> ---
>
> We sincerely appreciate your time, and we are glad to answer any additional questions you may have.

---

> > ### Comment · Reviewer_W8L4 · 2024-11-23
> >
> > I thank the authors' prompt response and apologize for my slightly late reply.
> >
> > I would like to comment on some points mentioned in your rebuttal.
> >
> > RW1: "in the real applications of regular binary classification tasks, the complexity of graph structures is typically less pronounced when compared to the intricate structures found in real-world graph anomaly detection tasks" This is not convincing, as there are a lot of node classification tasks where the graph topology is heavily used.
> >
> > RW1: "regular binary classification tasks usually do not suffer from the limited supervision problem in the real deployment. However, it is a ubiquitous problem within the area of graph anomaly detection tasks." this is not convincing as there are **many few-shot node classification efforts** such as [1].
> >
> > RW3.2 Thanks for your clarification; I think, in short, it is the addition of the output (probability vector).
> >
> > Overall, I still believe that this paper is not tailored for the few-shot graph anomaly detection task. For this reason, I prefer to maintain my borderline evaluation.
> >
> > [1] Zhou, Fan, et al. "Meta-gnn: On few-shot node classification in graph meta-learning." Proceedings of the 28th ACM International Conference on Information and Knowledge Management. 2019.

---

> > > ### Author Response · Authors · 2024-11-24
> > > **Author Response for Reviewer W8L4**
> > >
> > > Thanks for your detailed reading and insightful feedback on our rebuttal. Please kindly refer to our further response to address your concerns below.
> > >
> > > Although there are node classification tasks where the graph topology is **heavily used**, their design may not be **properly used** in the graph anomaly detection area. As we mentioned in Section 1, the complex structures, like circle-shaped structures in money laundry, and tree-like structures in malicious reviews detection, have easily been observed in the corresponding graph anomaly detection tasks and can be explained with physical meanings, whose information can be better captured by our proposed SpaceGNN. In contrast, even though previous works that focus on general node classification may leverage other topology properties, like heterophily and homophily, they can be demonstrated as inferior in real-world graph anomaly datasets as shown in our experiments. To be specific, we have compared many general node classification algorithms, like GCN, GraphSAGE, GAT, and GIN. As shown in Table 1, our SpaceGNN can consistently outperform them on all 9 real-world graph anomaly detection datasets. This shows that our model can better capture complex structures by using multiple spaces compared to previous works.
> > >
> > > As for the second comment “There are many few-shot node classification efforts”, we admit that there are a few node classification models designed for few-shot node classification efforts. However, the “Meta-GNN” mentioned in the comments focuses on a meta-learning setting, which is different from our graph anomaly detection setting. Specifically, in a meta-learning setting, the task aims to utilize a pre-trained general classification model trained on $|C_1|$ classes to classify nodes with few-shot samples within totally new $|C_2|$ classes, but in the graph anomaly detection tasks, we only have two classes, normal and abnormal, which both appear in the training set. Other than the work mentioned by the reviewer, the most commonly used technique to solve limited supervision settings is data augmentation. Such a technique might obtain promising performance when the data imbalance problem is not obvious in the dataset. Nevertheless, the ubiquitous imbalance problem in the graph anomaly detection datasets will hamper the effectiveness of the data augmentation technique, especially when there are only a few labeled data. As shown in Section 4.4 in our manuscript, the most recent graph anomaly model, CONSISGAD, tried to use a data augmentation technique to solve both data imbalance and limited supervision problems within graph anomaly detection datasets at the same time, but in our empirical demonstration, such a technique may not be effective enough. Thus, to better solve the limited supervision issue without being severely influenced by the imbalanced nature of graph anomaly detection datasets, we instead introduce a model augmentation way to collect enough information. Specifically, with independent spaces that can effectively handle different complex graph structures, our weighted soft-voting ensemble framework can leverage more comprehensive information to solve the graph anomaly detection tasks without being negatively influenced by the imbalance problem which mainly exists in the graph anomaly detection area. In summary, our proposed model is a solid trial to tackle these challenges, i.e., **complex but meaningful data structures, imbalance data distribution, and limited supervision**, that especially appear in real graph anomaly detection problems.
> > >
> > > Furthermore, although our proposed model is not designed for general node classification problems, as they might have similar issues to some extent, we remain positive that our SpaceGNN can perform well in general tasks, which can be left in future work. Also, we would appreciate it if the reviewer could provide more insights, like naming a few related general node classification datasets that contain all above mentioned three challenges in the graph anomaly detection area simultaneously, or frameworks designed for general node classification tasks but able to handle all above mentioned three challenges effectively. We would be happy to provide more experimental demonstrations to further address the concerns about the connection between our model and the graph anomaly detection tasks.
> > >
> > > We sincerely appreciate your time, and we are glad to answer any additional questions you may have.

---

> > > > ### Comment · Reviewer_W8L4 · 2024-11-28
> > > >
> > > > I thank the authors' response.
> > > >
> > > > After carefully reading the further response, I still believe this method is **not well-motivated**, especially considering that the abstract highlights that it is designed to address "extremely limited labels." I did not find any design **tailored for such an extremely few-shot setting**. The authors responded that to address this limited label problem, they adopt "a model augmentation way to collect enough information," which is vague and not convincing. I did not see clear evidence from Section 4.4 showing that the ensemble method can alleviate the few-shot limitation.
> > > >
> > > > Also, I saw other reviewers are satisfied with its experiments, and I do not find flaws in them, either. Hence, if other reviewers and AC believe that this part outweighs the motivation and design of the method, I would not be frustrated to see this paper being accepted, while I prefer to maintain my current evaluation.

---

> > > > > ### Author Response · Authors · 2024-11-30
> > > > > **Author Response for Reviewer W8L4**
> > > > >
> > > > > Thanks for your detailed reading and insightful feedback on our rebuttal. We also appreciate your admission of our contribution to the graph anomaly detection area. To further clarify the motivation and soundness of our proposed model, we provide more evidence and explanations below.
> > > > >
> > > > > According to the comments, we assume that the final concern would be that the reviewer thought the ensemble method was not tailored to alleviate the few-shot limitation. Hence, to further address this concern, we will explain our reasons from three aspects.
> > > > >
> > > > > First, as shown in Section 4.4, we start by analyzing the most recent work, CONSISGAD[1], which is tailored to deal with the limited supervision problem in the graph anomaly detection area. Specifically, they utilize a learning framework to perform data augmentation to mitigate limited supervision issues. However, as shown in our empirical analysis, such a framework is less effective in generating useful synthetic data when applied to graph anomaly detection datasets. This analysis motivates us to introduce a superior framework for tackling this issue without using synthetic information, which makes leveraging different views of data samples as data augmentation a natural choice. With several GNNs in different independent spaces, our framework can be effective in collecting more information as shown in our empirical and theoretical results, which proves that our ensemble framework can mitigate the limited information without bringing possible noises from generated data. Furthermore, as theoretically shown in Propositions 1 and 2, given the same dataset, the ensemble model would perform better than a single model, which also shows that the ensemble model can collect more information effectively.
> > > > >
> > > > > Second, not limited to the graph learning area, utilizing ensemble frameworks to deal with limited supervision problems is also prevalent in other areas. Previous works, such as SESoM[2], E3[3], ELMOS[4], and E3BM[5], can also support our motivation to use an ensemble framework to solve the few-shot limitation in various problems. For example, ELMOS claims and demonstrates that “we introduce ensemble learning in the first phase to improve the FSC (Few-shot Classification) performance”. Similar evidence can be also observed in other ensemble frameworks for different few-shot tasks. From this aspect, we can see leveraging an ensemble framework can be a reasonable and popular choice to alleviate the few-shot problem.
> > > > >
> > > > > Third, extensive experiments, as confirmed by the reviewers that they are satisfied with the results, show that our proposed ensemble framework can outperform single models under limited supervision settings to a large margin, which further demonstrates that, compared to single models, our SpaceGNN can be useful under such an environment.
> > > > >
> > > > > In conclusion, from the above aspects, we believe our ensemble framework can be used to alleviate the few-shot limitation.
> > > > >
> > > > > We sincerely appreciate your time, and we are glad to answer any additional questions you may have.
> > > > >
> > > > > Reference:
> > > > >
> > > > > 1. Nan Chen, Zemin Liu, Bryan Hooi, Bingsheng He, Rizal Fathony, Jun Hu, Jia Chen. Consistency Training with Learnable Data Augmentation for Graph Anomaly Detection with Limited Supervision. ICLR 2024.
> > > > >
> > > > > 2. Xiangyu Peng, Chen Xing, Prafulla Kumar Choubey, Chien-Sheng Wu, Caiming Xiong. Model ensemble instead of prompt fusion: a sample-specific knowledge transfer method for few-shot prompt tuning. ICLR 2024.
> > > > >
> > > > > 3. Aref Azizpour, Tai D. Nguyen, Manil Shrestha, Kaidi Xu, Edward Kim, Matthew C. Stamm. E3: Ensemble of Expert Embedders for Adapting Synthetic Image Detectors to New Generators Using Limited Data. CVPR Workshop 2024.
> > > > >
> > > > > 4. Sai Yang, Fan Liu, Delong Chen, Jun Zhou. Few-shot Classification via Ensemble Learning with Multi-Order Statistics. IJCAI 2023.
> > > > >
> > > > > 5. Yaoyao Liu, Bernt Schiele, Qianru Sun. An Ensemble of Epoch-Wise Empirical Bayes for Few-Shot Learning. ECCV 2020.

---

> > > > > > ### Author Response · Authors · 2024-12-03
> > > > > >
> > > > > > Dear Reviewer,
> > > > > >
> > > > > > We sincerely hope that we have addressed all your concerns, and we are glad to answer any additional questions you may have.
> > > > > >
> > > > > > Warm regards,
> > > > > >
> > > > > > The Authors of Submission6161

---

> ### Author Response · Authors · 2024-11-23
> **Author Response for Reviewer W8L4**
>
> Dear Reviewer,
>
> We sincerely hope that we have addressed all your concerns, and we are glad to answer any additional questions you may have.
>
> Warm regards,
>
> The Authors of Submission6161

---

### Official Review · Reviewer_XNDy · 2024-11-03

**Soundness:** 3
**Presentation:** 2
**Contribution:** 3
**Rating:** 6
**Confidence:** 4

**Summary:**

This paper aims to address two problems of existing methods in node anomaly detection, Euclidean space only and limited supervision. The authors propose SpaceGNN, consisting of Learnable Space Projection, Distance Aware Propagation, and Multiple Space Ensemble. Both theoretical analysis and experimental results demonstrate the effectiveness of the proposed methods.

**Strengths:**

- The paper proposes to adopt ensemble of multiple spaces with different curvature in the task of node anomaly detection, which is novel and effective.
- The paper offers empirical and theoretical analysis to illustrate the soundness of the proposed method.
- The paper provides experimental results to prove the effectiveness of the proposed method.

**Weaknesses:**

- The paper has no details on what $\kappa$ is like after learning.
- The Multiple Space Ensemble may increase the computational complexity. There is neither theoretical analysis nor efficiency experiments to show how much additional cost is compared to a single model.
- The authors have not released code for reproducibility.
- Although the paper is understandable, there are several minor presentation issues that reduce readability:
  - The captions of figures are usually at the bottom of the figure instead of on the top in ICLR format.
  - The sections in the Appendix are not necessarily subsections in a single Section A.
  - The abbreviation of Multiple Space Ensemble (MSE) may be confusing to the audience and can be misinterpreted as another commonly used term Mean Squared Error.
  - Line 281: avoid in-place operation to $H^l$
  - Line 313: $s_{ij}$ is not clear defined

**Questions:**

- GADBench adopts AUC, AUPRC, and Rec@K as metrics for comprehensive evaluation. Is there any particular reason you use AUC and F1 instead?
- GADBench also provides a semisupervised setting with limited labels. Instead of using a random split, it will be more convincing and reproducible to run experiments on the public split provided by GADBench.
- Line 292 claims $\kappa$ is learnable, but line 405 seems to restrict $\kappa$ either positive or negative.

---

> ### Author Response · Authors · 2024-11-19
> **Author Response for Reviewer XNDy (1/3)**
>
> We appreciate your comprehensive and constructive review. Your crucial comments on experiments are helpful for us to improve our manuscript. Our point-to-point responses to your comments are given below.
>
> ---
> **W1**: The paper has no details on what $\kappa$ is like after learning.
>
> **RW1**: Thanks for pointing that out, we have revised our manuscript and included Table 11 of learned $\kappa$ in Appendix I. Please kindly refer to our revised manuscript.
>
> ---
>
> **W2**: The Multiple Space Ensemble may increase the computational complexity. There is neither theoretical analysis nor efficiency experiments to show how much additional cost is compared to a single model.
>
> **RW2**: Thanks for pointing that out, we have revised our manuscript and included the theoretical analysis of time complexity in Appendix J. According to our analysis of the updated manuscript, the computational complexity will not increase compared to a single model, e.g. GAT, with proper hyperparameters. Please kindly refer to our revised manuscript.
>
> ---
>
> **W3**: The authors have not released code for reproducibility.
>
> **RW3**: Thanks for pointing that out, we have uploaded our source code for reproducibility, attached with detailed steps in README. Please kindly refer to the supplemental materials on OpenReview.
>
> ---
>
> **W4**: Although the paper is understandable, there are several minor presentation issues that reduce readability:
>
> The captions of figures are usually at the bottom of the figure instead of on the top in ICLR format.
>
> The sections in the Appendix are not necessarily subsections in a single Section A.
>
> The abbreviation of Multiple Space Ensemble (MSE) may be confusing to the audience and can be misinterpreted as another commonly used term Mean Squared Error.
>
> Line 281: avoid in-place operation to $H^l$
>
> Line 313: $s_{ij}$ is not clear defined
>
> **RW4**: Thanks for pointing that out, we have revised our manuscript based on your constructive suggestions. Please kindly refer to the latest version of our manuscript.

---

> ### Author Response · Authors · 2024-11-19
> **Author Response for Reviewer XNDy (2/3)**
>
> ---
>
> **Q1**: GADBench adopts AUC, AUPRC, and Rec@K as metrics for comprehensive evaluation. Is there any particular reason you use AUC and F1 instead?
>
> **RQ1**: There are three main reasons for using AUC and F1 as our performance metrics.
>
> First, in the field of supervised graph anomaly detection, researchers commonly adopt AUC with one of the metrics in F1 and AUPRC to evaluate the performance of their frameworks, as evidenced in previous studies [1, 2, 3, 4, 5]. Note that the authors of BWGNN [4], who are also the authors of GADBench, select AUC and F1 as evaluation metrics. Hence, in the literature, it is quite standard to adopt AUC and F1 as the metrics to show the effectiveness of our proposed model.
>
> In addition, F1 is the combination of precision and recall into one metric by calculating the harmonic mean between those two, which can reflect both AUPRC and Rec@K to some extent.
>
> Third, according to Appendix A in GADBench, the baseline model, XGBOD, is an unsupervised framework. It is hard to report a reasonable F1 for such unsupervised models, as calculating F1 requires selecting a specific threshold. This might be the reason why GADBench utilizes AUPRC and Rec@K as their metrics, which are the most commonly used evaluation metrics for unsupervised graph anomaly detection models [6, 7, 8].
>
> In conclusion, in the literature, F1 can provide a clear, interpretable, and balanced evaluation of precision and recall for a specific threshold selected based on the validation set, making it a practical and widely used metric in the area of supervised graph anomaly detection tasks, while AUPRC and Rec@K are more suitable for the unsupervised graph anomaly detection tasks.
>
> ---
>
> **Q2**: GADBench also provides a semisupervised setting with limited labels. Instead of using a random split, it will be more convincing and reproducible to run experiments on the public split provided by GADBench.
>
> **RQ2**: There are two main reasons for not using the data split in GADBench.
>
> First, while the authors of GADBench provided a *preprocess.ipynb* file for data split preprocessing in their official GitHub repository, the splits outlined in this file are not aligned with our focused settings. For instance, the training and validation ratio of Tolokers and Questions are 50% and 25%, respectively; those of T-Social, T-Finance, Reddit, and Weibo are 40% and 20%, respectively; and those of Amazon, YelpChi, and DGraph-Fin follow the original data split from their corresponding papers, which do not adhere to our limited supervision split criteria. Therefore, we cannot track the official GADBench split with limited labels.
>
> Second, the authors of GADBench only specify the training set size in the semi-supervised setting, leaving the validation set size unspecified. Consequently, in the absence of this crucial information, we are unable to determine the appropriate validation set size for such a scenario. According to the fully-supervised setting, the validation set size is set at 20%, this does not align with our objective of a limited supervision setting due to the substantial size of the validation set. Our study focuses specifically on **extreme cases where both training and validation data are severely constrained**. Specifically, as we mentioned in Section 5.1, we randomly divide each dataset into 50/50 for training/validation, with the remaining nodes reserved for testing. This methodology mirrors real-world graph anomaly detection tasks where limited training and validation data are common challenges.
>
> ---
>
> **Q3**: Line 292 claims $\kappa$ is learnable, but line 405 seems to restrict $\kappa$ either positive or negative.
>
> **RQ3**: As shown in Section 4.4 of our manuscript, we utilize an ensemble way to collect comprehensive information from Hyperbolic, Euclidean, and Spherical spaces. The reason why we restrict $\kappa$ either positive or negative is that we aim to employ several independent GNNs to learn from independent spaces, instead of mixing various spaces in different layers of each single GNN. The preservation of the independence of different spaces perfectly satisfies the requirement of ensemble methods, as shown in our theoretical analysis in Section 4.4. To sum up, we restrict each single GNN in a single space (one of Hyperbolic, Euclidean, and Spherical spaces) with learnable $\kappa$ to provide comprehensive independent information collected from independent spaces for ensembling.
>
> ---
>
> We sincerely appreciate your time, and we are glad to answer any additional questions you may have.

---

> ### Author Response · Authors · 2024-11-19
> **Author Response for Reviewer XNDy (3/3)**
>
> Reference:
>
> 1. Nan Chen, Zemin Liu, Bryan Hooi, Bingsheng He, Rizal Fathony, Jun Hu, Jia Chen. Consistency Training with Learnable Data Augmentation for Graph Anomaly Detection with Limited Supervision. ICLR 2024.
>
> 2. Yuan Gao, Xiang Wang, Xiangnan He, Zhenguang Liu, Huamin Feng, Yongdong Zhang. Addressing Heterophily in Graph Anomaly Detection: A Perspective of Graph Spectrum. WWW 2023.
>
> 3. Yuchen Wang, Jinghui Zhang, Zhengjie Huang, Weibin Li, Shikun Feng, Ziheng Ma, Yu Sun, Dianhai Yu, Fang Dong, Jiahui Jin, Beilun Wang, Junzhou Luo. Label Information Enhanced Fraud Detection against Low Homophily in Graphs. WWW 2023.
>
> 4. Jianheng Tang, Jiajin Li, Ziqi Gao, Jia Li. Rethinking Graph Neural Networks for Anomaly Detection. ICML 2022.
>
> 5. Ziwei Chai, Siqi You, Yang Yang, Shiliang Pu, Jiarong Xu, Haoyang Cai, Weihao Jiang. Can Abnormality be Detected by Graph Neural Networks? IJCAI 2022.
>
> 6. Jingyan Chen, Guanghui Zhu, Chunfeng Yuan, Yihua Huang. Boosting Graph Anomaly Detection with Adaptive Message Passing. ICLR 2024.
>
> 7. Hezhe Qiao, Guansong Pang. Truncated Affinity Maximization: One-class Homophily Modeling for Graph Anomaly Detection. NeurIPS 2023.
>
> 8. Dmitrii Gavrilev, Evgeny Burnaev. Anomaly Detection in Networks via Score-Based Generative Models. ICML 2023 Workshop SPIGM.

---

> ### Author Response · Authors · 2024-11-23
> **Author Response for Reviewer XNDy**
>
> Dear Reviewer,
>
> We sincerely hope that we have addressed all your concerns, and we are glad to answer any additional questions you may have.
>
> Warm regards,
>
> The Authors of Submission6161

---

> ### Comment · Area_Chair_ZytW · 2024-11-25
> **Please reply to the authors' response.**
>
> Dear reviewers,
>
> The ICLR author discussion phase is ending soon. Could you please review the authors' responses and take the necessary actions? Feel free to ask additional questions during the discussion. If the authors address your concerns, kindly acknowledge their response and update your assessment as appropriate.
>
>
> Best,
> AC

---

> ### Comment · Reviewer_XNDy · 2024-11-25
>
> I appreciate authors' response. Although some of my concerns have been addressed, I still cannot agree with some of authors' claims on evaluation:
>
> - Evaluation metrics: existing **supervised** benchmark GADBench adopts AUC, AUPRC, and Rec@K. It is not convincing to have a different metric.
> - Dataset split: the semi-supervised setting in GADBench has 20 positive labels (anomalous nodes) and 80 negative labels (normal nodes) for both the training set and the validation set in each dataset, which is not so different from 50/50 in authors' setting. The public split should be used.
>
> Therefore, I am going to keep my score for now and reserve my right adjust the score.

---

> > ### Author Response · Authors · 2024-11-27
> > **Author Response for Reviewer XNDy**
> >
> > Thanks for your detailed reading and insightful feedback on our rebuttal. Please kindly refer to our further response to address your concerns below.
> >
> > We admit that AUC, F1, AUPRC, and Rec@K are all reasonable metrics in the area of graph anomaly detection based on the recent benchmark papers, GADBench [1] and PyGOD [2]. Hence, to address the reviewer’s concern, we have emailed the authors of GADBench and confirmed the semi-supervised setting in their paper. Specifically, there are 20 anomalous nodes and 80 normal nodes for both the training and the validation set in each dataset. We add the new experiments in Appendix L in this setting and find that our proposed SpaceGNN can still consistently outperform all the baselines on almost all datasets using AUC, AUPRC, and Rec@K as our metrics.
> >
> > For instance, compared with the two most competitive models, XGBGraph and CONSISGAD, our SpaceGNN can take the lead up to 23.6% and 15.8% in terms of AUPRC, and 23.7% and 13.2% in terms of Rec@K,  on the largest dataset T-Social. As for other datasets, our proposed models can also exceed them by a considerable margin. In conclusion, these new experiments further demonstrate the effectiveness of our proposed model.
> >
> > We sincerely appreciate your time, and we are glad to answer any additional questions you may have.
> >
> > Reference:
> >
> > 1. Jianheng Tang, Fengrui Hua, Ziqi Gao, Peilin Zhao, Jia Li. GADBench: Revisiting and Benchmarking Supervised Graph Anomaly Detection. NeurIPS 2023.
> >
> > 2. Kay Liu, Yingtong Dou, Xueying Ding, Xiyang Hu, Ruitong Zhang, Hao Peng, Lichao Sun, Philip S. Yu. PyGOD: A Python Library for Graph Outlier Detection. Journal of Machine Learning Research 2024.

---

> ### Comment · Reviewer_XNDy · 2024-11-28
>
> I appreciate the authors' additional experiments. I will keep my score positive and suggest a weak accept (between 6 and 8) for this paper.

---

> > ### Author Response · Authors · 2024-12-03
> >
> > Dear Reviewer,
> >
> > Thank you very much for the time and effort you have put into reviewing our paper and providing insightful comments. Moreover, we are also delighted to know that you are satisfied with our responses and confirm our contribution to the graph anomaly detection area. We deeply appreciate your support for our work.
> >
> > Warm regards,
> >
> > The Authors of Submission6161

---

### Author Response · Authors · 2024-11-19
**Modifications in the revised version of our manuscript**

We express our gratitude to all the reviewers for their thorough and constructive feedback. Taking into consideration the valuable comments provided by the reviewers, we have incorporated the following modifications in our revised manuscript, which have been highlighted in blue to facilitate the reviewing process.

---

**Section 1**

- revise the abbreviation of the Multiple Space Ensemble. (Reviewer **XNDy**)

- move the title of the figure to the bottom. (Reviewer **XNDy**)

**Section 4**

- revise the abbreviation of the Multiple Space Ensemble. (Reviewer **XNDy**)

- move the title of the figures to the bottom. (Reviewer **XNDy**)

- revise the equations by avoiding the in-place operation. (Reviewer **XNDy**)

- revise Definition 2 by providing details of $s_{ij}^{\kappa}$. (Reviewer **XNDy**)

- transition Theorems 3 and 4 to Propositions 1 and 2. (Reviewer **W8L4**)

**Appendix A**

- transition Theorems 3 and 4 with Propositions 1 and 2. (Reviewer **W8L4**)

**Appendix C**

- revise Algorithm 3 by avoiding the in-place operation. (Reviewer **XNDy**)

**Appendix E**

- move the title of the figure to the bottom. (Reviewer **XNDy**)

**Appendix I**

- add Table 11 to show the learned kappa. (Reviewer **XNDy**)

**Appendix J**

- add time complexity analysis of SpaceGNN. (Reviewer **XNDy**)

**Appendix K**

- add figures to show the superior performance of SpaceGNN varying the size of the training set. (Reviewer **Snna**)

---

---

### Meta-Review · Area_Chair_ZytW · 2024-12-18

**Metareview:**

The paper presents an approach to node anomaly detection by leveraging multiple spaces for node representation and ensemble learning. The authors introduce a Learnable Space Projection function and a Multiple Space Ensemble module, demonstrating improvements over existing methods in both theoretical analysis and empirical results.

The strengths of the paper include its innovative use of multiple spaces, comprehensive experimental validation across nine datasets, and the ability to handle extremely limited supervision effectively. However, some concerns were raised about the increased computational complexity and the clarity of certain theoretical aspects. The authors addressed these concerns through detailed responses and additional experiments, which satisfied most reviewers. Despite these issues, the paper seems to provide a well-motivated solution to graph anomaly detection. Therefore, I recommend accepting this paper for publication

**Additional Comments On Reviewer Discussion:**

The authors addressed most concerns through detailed responses and additional experiments, which satisfied most reviewers.

---

### Decision · Program_Chairs · 2025-01-22

Accept (Poster)